# Skip a Layer or Loop It? Learning Program-of-Layers in LLMs

**Ziyue Li** [1]   **Yang Li**   **Tianyi Zhou** [2]

Project: https://github.com/tianyi-lab/PoLar

## Abstract

Large language models (LLMs) perform inference by following a fixed depth and order, non-recurrent execution of all layers. We reveal the wide existence of training-free, flexible, dynamic "program-of-layers (POLAR)", where pretrained layers can be packed as modules and then skipped or looped to form a customized program for each input. For most inputs, substantially shorter program executions can achieve the same or better accuracy, while incorrect predictions of the original LLM can be corrected by alternative programs with fewer layers. These observations indicate that inference admits multiple valid latent computations beyond the standard forward pass. To efficiently achieve POLAR in practice, we propose a lightweight POLAR prediction network, which learns to generate execution programs that dynamically skip or repeat pretrained layers for each input. Experiments on mathematical reasoning benchmarks demonstrate that POLAR consistently improves accuracy over standard inference and prior dynamic-depth methods, often while executing fewer layers, and that these gains persist under out-of-distribution evaluation. Our results suggest that fixed-depth execution captures only a narrow subset of an LLM's latent reasoning capacity.

## 1. Introduction

Generalist foundation models, e.g., LLMs and VLMs, uniformly deploy a static, pre-defined architecture to all inputs, despite their diversity and high variance in complexity and difficulty (Liu et al., 2020; Xin et al., 2020; Zhou et al., 2020; Liu et al., 2021a). In contrast, conventional problem

[1]University of Maryland, College Park, MD, USA [2]MBZUAI, Abu Dhabi, UAE. Correspondence to: Tianyi Zhou <tianyi.zhou@mbzuai.ac.ae>.

*Proceedings of the 43rd International Conference on Machine Learning*, Seoul, South Korea. PMLR 306, 2026. Copyright 2026 by the author(s).

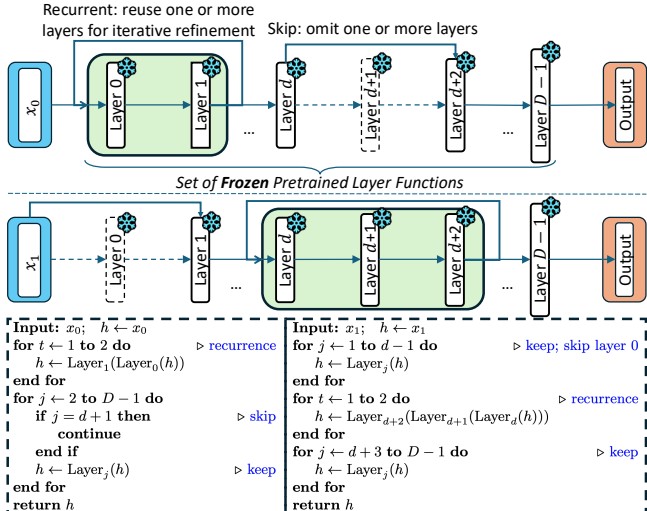

*Figure 1.* **Program-of-layers (POLAR) for two different inputs.** The $D$ layers in a pretrained LLM define $D$ functions $f_0, \ldots, f_{D-1}$. Instead of calling them in a static fixed order from $f_0$ to $f_{D-1}$, the dynamic inference of POLAR executes an *input-specific program* $\pi = (i_1, \ldots, i_K)$ that calls the functions with layer *skipping* and *recurrence*. POLAR enables a training-free architecture of dynamic depth for different inputs, yielding diverse latent computations that cannot be fully covered by existing methods.

solving by programs can be more flexible and adaptive in algorithmic structures and complexity. For example, an experienced programmer can save more steps and compute on easier tasks, and meanwhile knows how to scale up the space/time complexity to address more challenging problems. However, these programs are specifically designed and optimized for every problem class, so they are not as general as LLMs. This raises the questions: *Is it always optimal and efficient to apply the same architecture or "program", i.e., forward pass through all the layers in a fixed order, to different tasks? Can a generalist model further optimize its "program" applied to each input?*

In this paper, we formulate layers in a pretrained LLM as a library of atomic functions that a program can call in arbitrary order for arbitrary times. This formulation allows us to represent a dynamic model architecture during inference as a program-of-layers (POLAR) for each input, as

illustrated in Figure 1. As the first empirical study of its kind, we investigate POLAR beyond the standard forward pass by Monte-Carlo Tree Search (MCTS) and find that better (more accurate and/or shorter) programs almost always exist for every input task evaluated. Unlike previous works on layer-skipping/recurrence, early exit, and looped transformer (Liu et al., 2020; Xin et al., 2020; Zhou et al., 2020; Fan et al., 2019; 2024; Yang et al., 2023), which only adopt one operation (either skip or repeat) to produce architectures of dynamic depths, our empirical study on the MCTS-searched programs reveals that searching in a joint space of layer-skip/repeat often discovers much better programs than those found in separate spaces. While most effective programs can be shorter than the default, increasing program complexity via skip/repeat operations can substantially improve the output quality, especially on more difficult tasks. In addition, most successful programs are predominantly composed of contiguous layer segments. These observations not only verify the broad existence of better POLAR without requiring any training, but also motivate a practical POLAR prediction method that avoids the expensive cost of MCTS in POLAR's large search space. In particular, we aim to replace search-based program discovery with a direct, inference-time mechanism for generating execution programs. Instead of enumerating or exploring execution paths for each input (Li et al., 2025), our goal is to predict an input-specific program-of-layers that determines how pretrained layers are executed during inference. This shifts program selection from an online search problem to a single-shot prediction problem, enabling practical deployment of program-of-layers inference in LLMs.

To this end, we propose a POLAR algorithm that predicts execution programs over frozen pretrained layers at inference time. The predicted execution program specifies how pretrained layers are selectively skipped or recurrently applied and is executed once to produce the final output. This design offers several advantages. First, it makes program-of-layers inference computationally feasible by eliminating the need for expensive per-input search. Second, by jointly supporting layer skipping and recurrence within a unified execution framework, POLAR strictly generalizes prior dynamic-depth methods that are limited to a single form of execution control. Third, it enables flexible test-time computation scaling in fully frozen models, allowing inference to adapt to input difficulty while preserving model generality.

We evaluate POLAR on a range of mathematical reasoning benchmarks using multiple pretrained LLMs. Our results show that POLAR consistently improves accuracy over standard inference and prior dynamic-depth methods, often while executing fewer layers on average. Moreover, increasing the number of candidate execution programs yields strong test-time computation scaling, and execution programs learned on in-distribution data generalize effectively

to out-of-distribution benchmarks across diverse domains.

## 2. Dynamic Inference as a Program-of-Layers (POLAR) in Large Language Models

Inference in pretrained LLMs is implemented as a fixed-depth, fixed-order forward pass: every input is processed by executing the same sequence of transformer layers. Yet inputs to LLMs vary dramatically in difficulty. Some are answered correctly with minimal reasoning, while others require complex, multi-step computation. This discrepancy raises a basic question: **Is the standard forward pass sufficient for correct inference across diverse inputs?**

One possibility is that this fixed computation is indeed sufficient for all cases. Another is that correct prediction requires input-dependent variation in computation. In this work, we investigate the latter possibility. Such variation can occur either in token space, through longer and more explicit chains of thought, or within the model's hidden states, a form of computation we refer to as *latent reasoning*.

> **Conjecture: Inference as Program-of-Layers**
>
> *Define the layers in a pretrained LLM as functions, for each input, there can exist multiple distinct executions of programs-of-layers (beyond the standard forward pass) that produce correct predictions.*

**Inference as the execution of a *program*.** In this view, inference is a step-by-step procedure that selects and composes pretrained modules. The execution may vary across inputs in both length and order, while each module remains a fixed, pretrained function.

Consider a pretrained LLM with $D$ transformer layers, where each layer defines a fixed computation function

$$f_i : \mathbb{R}^{T \times d} \to \mathbb{R}^{T \times d}, \quad i \in \{0, \dots, D-1\}.$$

A program is defined as a finite sequence of layer indices

$$\pi = (i_1, i_2, \dots, i_K), \quad i_k \in \{0, \dots, D-1\},$$

which induces the composed computation

$$F_\pi = f_{i_K} \circ \cdots \circ f_{i_1}.$$

Executing a program applies this composition to the input and produces a prediction. A program is considered *valid* if it yields a correct prediction for a given input.

**Searching for valid execution programs.** We explore the space of execution programs using MCTS. Execution programs are variable-length sequences over pretrained transformer layers, allowing both skipping and repetition. This space is large, discrete, and highly non-convex, making exhaustive search infeasible. MCTS provides a principled way to prioritize promising partial programs, enabling us to verify the existence of valid programs and analyze their structural properties. We use MCTS strictly as a diagnostic

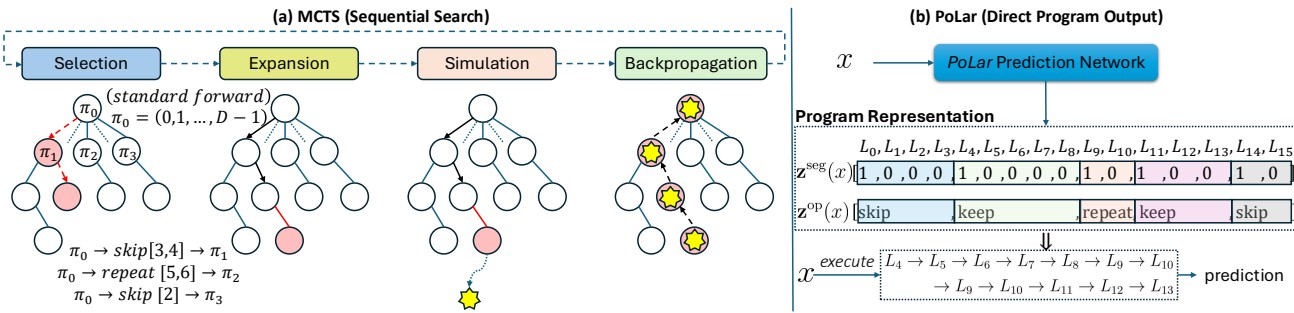

*Figure 2.* **Sequential MCTS (left) vs. End-to-end POLAR network (right) for prediction of programs.** (a) MCTS in the space of execution programs via sequential iterations of selection, expansion, simulation, and backpropagation. Each node represents a partial or complete execution program, and skip/repeat operations expand the search tree iteratively. This explicit and thorough search is expensive and impractical. (b) Our POLAR trains an end-to-end, lightweight prediction network that directly produces a program representation composed of (i) a binary mask $\mathbf{z}^{seg}(x)$ segmenting layers into modules, and (ii) a vector of operation labels $\mathbf{z}^{op}(x)$ that applies one operation out of *skip*, *keep*, or *repeat* to each module. Our method is scalable in practice and does not require sequential search.

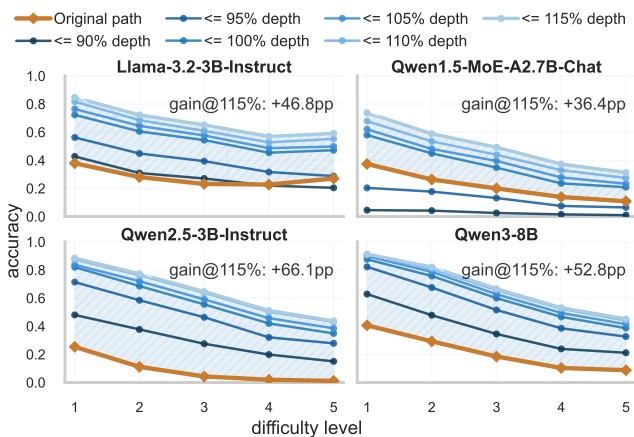

*Figure 3.* **Accuracy of MCTS discovered programs under varying execution-depth budgets** across five difficulty levels in DART-Math. We compare the original forward pass (orange) with 90–115% depth-budgeted programs (blue). Shaded regions denote the maximum gain achieved under the highest budget (115%).

tool rather than a practical inference-time method; implementation details are given in Appendix B. All experiments are conducted on **DART-Math** (Tong et al., 2024), a structured mathematical reasoning benchmark with five difficulty levels (DM-1 to DM-5). We evaluate four pretrained transformer models: *LLaMA-3.2-3B-Instruct*, *Qwen1.5-MoE-A2.7B-Chat*, *Qwen2.5-3B-Instruct*, and *Qwen3-8B*.

We summarize our empirical findings below.

---
**Finding 1**

*Layer recurrence-only performs better than layer skipping-only, but combining the two complementary operators produces the best program-of-layers.*

---

As shown in Table 1, execution programs that allow layer recurrence (**Loop**) consistently outperform those that allow

only layer skipping (**Skip**) across all evaluated models and difficulty levels. Moreover, combining skipping with recurrence (**Skip&Loop**) yields substantially larger gains than either operation alone, achieving the highest accuracy in every setting reported in Table 1. These results indicate that recurrence provides a stronger mechanism for improving inference than skipping alone, while the two operations play complementary roles when combined.

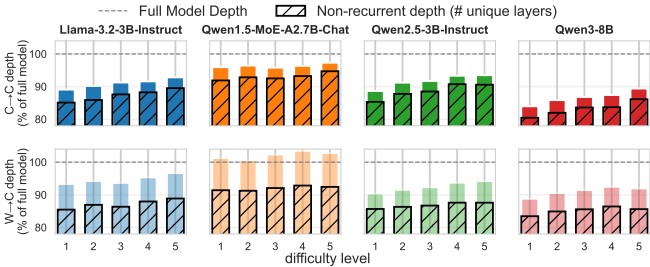

*Figure 4.* **Latent execution programs often admit shorter valid solutions.** We compare the standard forward-pass depth with that of MCTS-discovered valid programs, for initially correct (C→C) and initially incorrect (W→C) inputs. Bars report total execution depth as a fraction of full model depth, with hatched overlays indicating effective depth (the number of unique layers).

---
**Finding 2**

*(Occam's razor) Most valid execution programs are often shorter than the standard forward pass.*

---

As shown in Figure 3, which reports the best accuracy obtained by MCTS under explicit budgets on total layer executions, many inputs remain solvable even when the overall computation is constrained to be significantly shorter than the standard forward pass. Consistent with this trend, Figure 4 shows that across models we frequently discover valid execution programs that require fewer layer applications than standard inference. In particular, among inputs already

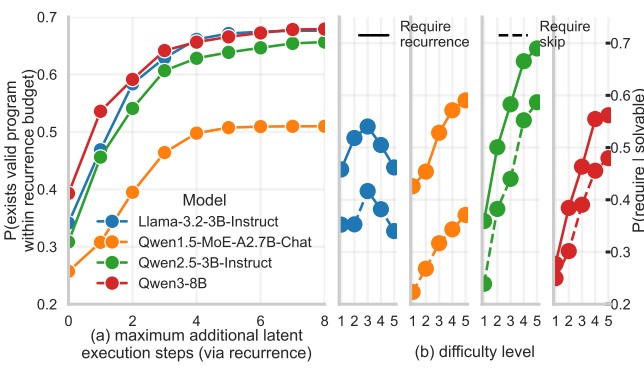

*Figure 5.* (a) **Test-time scaling via recurrence over layer segments.** Allowing more latent execution steps through segment recurrence leads to a monotonic increase in the probability of discovering valid execution programs across models. (b) **Recurrence and skipping are increasingly demanded for harder inputs.** The fraction of inputs relying on layer recurrence or skipping to be solved increases with increasing difficulty for most models, except LLaMA-3.2-3B-Instruct, whose deviation is explained by effective difficulty in Table 1.

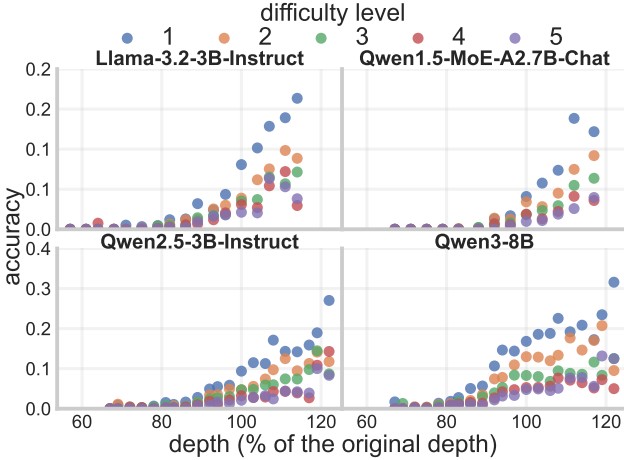

*Figure 6.* **Accuracy vs. total layer executions.** For each model, we report how the average accuracy of valid execution programs changes with the total number of layer executions (% of base model depth). Across models and difficulty levels, accuracy increases with executed layers, revealing a consistent effect of depth-scaling.

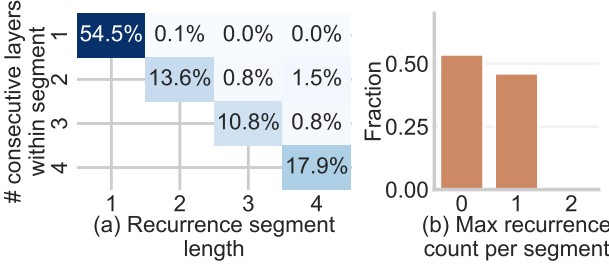

*Figure 7.* **Structural bias of valid execution programs.** Valid programs rely primarily on contiguous layer segments as modules (a) and require at most one recurrence of each module (b).

solved correctly by standard inference (C→C), 75.5% admit shorter valid programs. Even for inputs initially solved incorrectly (W→C), 36.2% admit shorter programs that correct the model's prediction. These results indicate that standard inference often over-computes, and that correct inference can frequently be achieved with substantially fewer latent computation steps.

> **Finding 3**
>
> *Increasing latent execution complexity systematically expands the space of valid programs and improves inference on harder inputs.*

While simple inputs often admit short execution programs (Finding 2), harder inputs demand greater test-time execution complexity. Across models and datasets, increasing execution depth and structural flexibility systematically expands valid program space and improves inference accuracy.

**(a) Test-time scaling expands the space of valid execution programs for latent reasoning.** As shown in Figure 5(a), allocating more test-time computation—via recurrence—monotonically increases the existence of valid execution programs across models. This establishes test-time scaling at latent reasoning: greater computation yields a larger feasible program space and higher correctness.

**(b) Harder inputs require more complex execution programs.** Figure 5(b) shows that the fraction of solvable inputs that require recurrence and/or skipping generally increases with dataset difficulty for most models. As task difficulty grows, valid execution programs become more constrained and increasingly rely on non-trivial execution structures rather than standard inference. This indicates that

higher latent execution complexity is not merely helpful, but often necessary for solving harder inputs. The different trend observed for LLaMA-3.2-3B-Instruct is explained by a mismatch between dataset-defined difficulty and the model's effective difficulty (Table 1).

**(c) Inference accuracy improves systematically with execution depth.** As shown in Figure 6, for all models and difficulty levels, the average accuracy of valid execution programs increases with total execution depth, measured relative to the original forward-pass depth. This reveals a consistent computation–accuracy trade-off: while many inputs admit short execution programs (Finding 2), harder inputs benefit from—and often require— deeper or recurrent execution to achieve correct inference.

Latent execution programs reveal a continuum of test-time inference behaviors that standard inference cannot access. Allocating more execution complexity enables harder inputs to be solved and yields higher accuracy.

*Table 1.* Accuracy (%) of programs searched for DART-Math (Diff 1–5) in different spaces: Base (standard forward), Skip (layer skipping), Loop (layer recurrence), and Skip&Loop (skipping + recurrence). **Gain** reports the absolute gain of **Skip&Loop** over Base. **Bold** indicates the best result, and underline indicates the second-best result in each row.

| Metric | Base | Skip | Loop | Skip&Loop | Gain |
|--------|------|------|------|-----------|------|
| **LLaMA-3.2-3B-Instruct** | | | | | |
| DM-1 | 37.9 | 45.7 | 54.9 | **84.7** | **+46.8** |
| DM-2 | 28.1 | 34.8 | 46.8 | **72.3** | **+44.2** |
| DM-3 | 23.2 | 29.7 | 38.0 | **65.2** | **+42.0** |
| DM-4 | 22.8 | 28.2 | 35.2 | **57.0** | **+34.2** |
| DM-5 | 27.1 | 31.7 | 39.0 | **59.1** | **+32.0** |
| **Qwen1.5-MoE-A2.7B-Chat** | | | | | |
| DM-1 | 37.4 | 42.7 | 57.7 | **73.2** | **+35.8** |
| DM-2 | 26.4 | 32.6 | 43.7 | **59.7** | **+33.3** |
| DM-3 | 20.0 | 23.8 | 34.4 | **50.4** | **+30.4** |
| DM-4 | 13.9 | 16.4 | 25.1 | **38.2** | **+24.3** |
| DM-5 | 10.9 | 13.3 | 20.4 | **32.4** | **+21.5** |
| **Qwen2.5-3B-Instruct** | | | | | |
| DM-1 | 25.4 | 47.0 | 60.2 | **87.4** | **+62.0** |
| DM-2 | 11.2 | 33.6 | 44.3 | **76.5** | **+65.3** |
| DM-3 | 4.3 | 25.1 | 35.5 | **65.0** | **+60.7** |
| DM-4 | 2.0 | 15.8 | 22.2 | **51.2** | **+49.2** |
| DM-5 | 1.2 | 13.2 | 18.1 | **44.5** | **+43.3** |
| **Qwen3-8B** | | | | | |
| DM-1 | 40.7 | 66.0 | 68.5 | **91.3** | **+50.6** |
| DM-2 | 29.3 | 50.6 | 57.4 | **82.2** | **+52.9** |
| DM-3 | 18.5 | 36.0 | 40.9 | **67.1** | **+48.6** |
| DM-4 | 10.4 | 23.8 | 29.2 | **53.6** | **+43.2** |
| DM-5 | 8.8 | 20.0 | 23.8 | **45.7** | **+36.9** |

---

> **Finding 4**
>
> *Valid execution programs are predominantly composed of contiguous layer segments and typically require at most a single recurrence per segment.*

As shown in Figure 7, valid execution programs discovered by pretrained models exhibit a strong structural bias toward simplicity. A segment denotes a set of layers executed as a unit and need not be contiguous, while recurrence always corresponds to re-execution of the same segment. We therefore analyze segment structure by measuring the number of consecutive layers within each segment. Most valid programs are dominated by highly local segments and involve at most a single recurrence; long-range jumps and deep iterative reuse are rare. Figure 7(a) shows that 54.5% of segments consist of a single layer, and over two-thirds contain at most two consecutive layers, whereas segments with predominantly non-consecutive layers account for less than 3.2% of cases. Consistently, Figure 7(b) indicates that most segments are repeated at most once. Together, these results reveal an inherent limitation of pretrained models as execution-program generators: their training objectives favor short-range, local reuse over rich program composition

and complex control flow.

These findings show that standard inference selects only one execution from a vast space of valid latent programs. While MCTS reveals this space, its reliance on sequential search over an exponentially large program space makes it impractical for inference. This motivates a different approach: rather than searching over programs at test time, we ask whether a lightweight model can directly predict execution programs. Figure 2 contrasts the MCTS-based sequential search with our proposed direct program prediction approach. In the remainder of this work, we pursue this learning-based alternative, retaining the benefits of latent program selection uncovered by MCTS while eliminating sequential search.

# 3. Learning Program-of-Layers (POLAR) in Large Language Models

Building on our empirical analysis (Section 2), we propose POLAR, a method for *programming* pretrained language models at inference time by predicting input-specific execution programs (Figure 2). POLAR dynamically segments and composes pretrained layers into reusable modules, enabling flexible computation without parameter updates.

## 3.1. Program Representation

We instantiate the function library using *packed modules*, which segment contiguous pretrained transformer layers into reusable computation units. For a pretrained model of depth $D$, an execution program specifies (i) a segmentation of layers into modules and (ii) an operation applied to each segment. Each execution program is represented by two discrete structures: a binary boundary mask encoding the segmentation, and an operation label vector specifying the segment-level operations.

**Segmentation.** We partition the $D$ layers of a pretrained model into contiguous segments

$$[0 = s_1, s_2), [s_2, s_3), \ldots, [s_M, s_{M+1} = D),$$

with each segment length bounded by $s_{j+1} - s_j \leq K_{\max}$. Segmentation is represented by a binary boundary mask

$$\mathbf{z}^{\text{seg}}(x) \in \{0, 1\}^D,$$

where $\mathbf{z}_i^{\text{seg}} = 1$ indicates that layer index $i$ starts a new segment, and $\mathbf{z}_i^{\text{seg}} = 0$ otherwise.

We set $K_{\max} = 4$ based on empirical evidence. **Finding 4** in Section 2 shows that valid execution programs are dominated by short, contiguous layer segments. Bounding the segment length therefore captures the dominant local execution structures while substantially reducing the complexity of the program space. Although this representation restricts the set of admissible programs, it preserves the most prevalent compositional patterns in practice and enables stable learning with strong empirical performance.

**Operations.** For each segment $[s_j, s_{j+1})$, the execution program assigns one of three operations $\{\mathsf{skip}, \mathsf{keep}, \mathsf{repeat}\}$, which determines how the segment is executed:

$$\mathsf{skip} : \emptyset,$$
$$\mathsf{keep} : [s_j, \ldots, s_{j+1} - 1],$$
$$\mathsf{repeat} : [s_j, \ldots, s_{j+1} - 1, \; s_j, \ldots, s_{j+1} - 1].$$

The $\mathsf{skip}$ operator omits a segment to reduce computation, while $\mathsf{repeat}$ applies a single additional pass. Operations are represented by a categorical label vector

$$\mathbf{z}^{\mathrm{op}}(x) \in \{\mathsf{skip}, \mathsf{keep}, \mathsf{repeat}\}^D,$$

where $\mathbf{z}_i^{\mathrm{op}}$ is defined only when $\mathbf{z}_i^{\mathrm{seg}} = 1$ (i.e., at segment start positions); labels at all other positions are ignored.

This operator set is intentionally minimal and empirically grounded. **Finding 4** in Section 2 shows valid execution programs rarely require more than a single re-execution within a segment, and that $\mathsf{skip}$ and $\mathsf{repeat}$ account for the most effective execution patterns, offering strong performance–efficiency trade-offs. The $\mathsf{keep}$ operator preserves the original computation when no modification is needed. Although our implementation allows at most one additional execution through $\mathsf{repeat}$, the representation is not fundamentally limited to a single recurrence. The operation vocabulary can be extended to $\{\mathsf{repeat}\text{-}2, \ldots, \mathsf{repeat}\text{-}k\}$ to support multiple recurrences per segment. We use a single-repeat operator because the MCTS traces in Section 2 show that effective programs rarely benefit from deeper repeated execution of the same segment. This choice keeps the prediction space tractable while covering the dominant valid programs.

## 3.2. Program-of-Layers (POLAR) Prediction Network

We train a lightweight predictor to output logits for the program representation defined in Section 3.1.

**Architecture.** Given an input $x$, we first encode it using a frozen embedding model (`Qwen3-Embedding-0.6B`), as token-level representations $\mathbf{H} = E(x) \in \mathbb{R}^{T \times d_q}$, where $T$ is the token length and $d_q$ is the hidden size of the embedding model. We project token representations to a working dimension $d$: $\tilde{\mathbf{H}} = \mathbf{H}\mathbf{W}_h \in \mathbb{R}^{T \times d}$.

*Layer queries.* We associate each pretrained transformer layer index $i \in \{0, \ldots, D-1\}$ with a learnable embedding $\mathbf{e}_i \in \mathbb{R}^d$, and stack them as $\mathbf{E} \in \mathbb{R}^{D \times d}$. These embeddings act as layer-specific queries.

*Cross-attention.* We apply multi-head cross-attention with layer embeddings as queries and token embeddings as keys/values: $\mathbf{X} = \mathrm{MHA}(\mathbf{Q}, \mathbf{K}, \mathbf{V}), \mathbf{Q} = \mathbf{E}, \; \mathbf{K} = \tilde{\mathbf{H}}, \; \mathbf{V} = \tilde{\mathbf{H}}$, where padding tokens are masked using the input attention mask. The output $\mathbf{X} \in \mathbb{R}^{D \times d}$ provides an input-conditioned representation for each layer index.

*Cross-layer encoder.* To model dependencies across model depth, we apply a lightweight transformer encoder over the layer dimension:

$$\mathbf{X}' = \mathrm{ENC}_{\mathrm{layer}}(\mathbf{X}) \in \mathbb{R}^{D \times d}.$$

This enables self-attention across layers, allowing decisions at each layer to depend on global depth context.

*Prediction heads.* Two linear heads produce logits for segmentation boundaries and operations:
$\boldsymbol{\ell}^{\mathrm{seg}} = \mathbf{X}'\mathbf{W}_{\mathrm{seg}} + \mathbf{b}_{\mathrm{seg}} \in \mathbb{R}^D, \boldsymbol{\ell}^{\mathrm{op}} = \mathbf{X}'\mathbf{W}_{\mathrm{op}} + \mathbf{b}_{\mathrm{op}} \in \mathbb{R}^{D \times 3}.$

**Supervision from Valid Execution Programs.** We supervise training using valid execution programs collected offline via MCTS (Section 2). Each program is deterministically parsed into program representation, producing ground-truth segmentation and operation labels $\mathbf{z}^{\mathrm{seg}}(x)$ and $\mathbf{z}^{\mathrm{op}}(x)$ in the format defined in Section 3.1. When multiple valid programs are available for an input and at least one is shorter than the full model depth, we down-weight the loss of the full-depth execution. This choice follows **Finding 2**, which shows that shorter valid programs are preferred while still preserving supervision from the original computation.

**Training Objective.** We train the predictor to match the ground-truth execution program, specified by segmentation and operation labels $(\mathbf{z}^{\mathrm{seg}*}(x), \mathbf{z}^{\mathrm{op}*}(x))$. Let $p_i^{\mathrm{seg}} = \sigma(\ell_i^{\mathrm{seg}})$ and $\mathbf{p}_i^{\mathrm{op}} = \mathrm{SOFTMAX}(\boldsymbol{\ell}_i^{\mathrm{op}})$. Segmentation is supervised with binary cross-entropy over boundary indicators:
$\mathcal{L}_{\mathrm{seg}} = -\sum_{i=0}^{D-1} \left[ \mathbf{z}_i^{\mathrm{seg}*} \log p_i^{\mathrm{seg}} + (1 - \mathbf{z}_i^{\mathrm{seg}*}) \log(1 - p_i^{\mathrm{seg}}) \right].$
Operation prediction uses a masked cross-entropy applied only at segment start positions. With mask $m_i = \mathbf{z}_i^{\mathrm{seg}*}$, we compute
$$\mathcal{L}_{\mathrm{op}} = -\sum_{i=0}^{D-1} m_i \cdot \log \mathbf{p}_i^{\mathrm{op}}[\mathbf{z}_i^{\mathrm{op}*}].$$
The final objective is $\mathcal{L} = \mathcal{L}_{\mathrm{seg}} + \mathcal{L}_{\mathrm{op}}$.

**Inference-Time Program Decoding.** At inference time, execution programs are decoded in two stages. First, segment boundaries are determined deterministically by thresholding the predicted segmentation logits $\boldsymbol{\ell}^{\mathrm{seg}}$. If any resulting segment exceeds the maximum length constraint $K_{\max}$, additional boundaries are inserted to enforce it, yielding segment start positions $\{s_j\}$. Conditioned on this segmentation, we compute operation log-probabilities at each segment start from the predicted logits:

$$\log p(o_j \mid x, s_j) = \log \mathrm{SOFTMAX}(\boldsymbol{\ell}_{s_j}^{\mathrm{op}})[o_j].$$

Rather than selecting operations independently via local argmax, we apply a small beam search over segment-level operation choices to account for non-local interactions between segments and to ensure globally consistent execution programs. This search operates over a highly constrained space and produces a ranked set of candidate execution programs $\pi(x)$. Finally, each candidate program is mapped deterministically to a concrete executed program using the segment-to-path rules in Section 3.1.

*Table 2.* **Pass@k accuracy under different inference strategies applied to LLaMA-3.2-3B-Instruct.** For difficulty levels DM-1 to DM-5, POLAR consistently outperforms standard inference and dynamic-depth baselines across $k$ values, where $\tau = 0$ denotes zero temperature.

| Method | p@ | DM-1 | DM-2 | DM-3 | DM-4 | DM-5 |
|---|---|---|---|---|---|---|
| **LLaMA-3.2-3B-Instruct** | | | | | | |
| Base ($\tau$=0) | 1 | 42.4 | 28.6 | 27.2 | 27.6 | 28.6 |
| Base (sampling) | 1 | 40.6 | 28.6 | 27.4 | 28.0 | 29.2 |
| | 2 | 44.0 | 35.0 | 29.6 | 30.4 | 32.8 |
| | 3 | 46.2 | 38.8 | 31.2 | 31.4 | 34.4 |
| | 4 | 47.0 | 41.4 | 32.2 | 32.2 | 34.8 |
| | 5 | 47.6 | 43.2 | 32.8 | 32.8 | 35.6 |
| ShortGPT | 1 | 3.2 | 2.6 | 4.6 | 3.0 | 3.8 |
| | 2 | 7.0 | 4.2 | 8.0 | 6.8 | 7.4 |
| | 3 | 10.6 | 8.8 | 9.8 | 9.4 | 10.8 |
| | 4 | 12.8 | 11.0 | 12.0 | 12.2 | 13.8 |
| | 5 | 16.4 | 14.6 | 14.0 | 15.2 | 17.0 |
| MindSkip | 1 | 6.8 | 6.4 | 6.4 | 4.2 | 6.2 |
| | 2 | 14.6 | 11.8 | 13.2 | 8.4 | 9.8 |
| | 3 | 22.8 | 18.8 | 18.8 | 12.8 | 15.4 |
| | 4 | 28.4 | 25.0 | 22.4 | 18.0 | 19.8 |
| | 5 | 35.2 | 30.0 | 27.6 | 22.2 | 21.8 |
| FlexiDepth | 1 | 9.0 | 10.0 | 7.2 | 3.4 | 2.6 |
| | 2 | 16.4 | 15.6 | 14.8 | 5.6 | 6.6 |
| | 3 | 20.0 | 21.4 | 19.8 | 7.2 | 9.4 |
| | 4 | 24.6 | 25.6 | 23.6 | 8.8 | 12.0 |
| | 5 | 28.6 | 29.2 | 26.4 | 10.0 | 13.8 |
| DR.LLM | 1 | 41.6 | 28.2 | 27.0 | 27.4 | 28.4 |
| | 2 | 46.2 | 32.8 | 31.8 | 29.4 | 31.8 |
| | 3 | 49.4 | 36.0 | 35.4 | 31.4 | 34.0 |
| | 4 | 50.8 | 38.8 | 36.8 | 32.8 | 35.0 |
| | 5 | 53.6 | 40.0 | 37.8 | 33.4 | 36.8 |
| **POLAR** | 1 | 46.2 | 30.2 | 28.2 | 28.8 | 30.2 |
| | 2 | 56.6 | 37.4 | 34.8 | 32.8 | 36.6 |
| | 3 | 62.8 | 42.8 | 39.4 | 35.6 | 40.2 |
| | 4 | 66.8 | 45.6 | 42.6 | 38.0 | 42.8 |
| | 5 | **68.4** | **48.0** | **46.0** | **40.4** | **45.8** |
| $\Delta$ vs. Base (sampling) | 5 | **+20.8** | **+4.8** | **+13.2** | **+7.6** | **+10.2** |

## 4. Experiments

We evaluate POLAR across both in-distribution and out-of-distribution benchmarks to assess whether learning *latent execution programs* provides a practical and transferable alternative to search-based test-time computation.

### 4.1. Experimental Setup

**Models.** We evaluate POLAR on a diverse set of pretrained, instruction-tuned LLMs spanning different architectures and scales: *LLaMA-3.2-3B-Instruct*, *Qwen1.5-MoE-A2.7B-Chat*, *Qwen2.5-3B-Instruct*, and *Qwen3-8B*. All models are used in a fully frozen setting with no parameter updates.

**Datasets.** We use **DART-Math** (Tong et al., 2024), a structured mathematical reasoning dataset with five difficulty levels (DM-1 to DM-5), as in-distribution benchmark. For out-of-distribution (OOD) evaluation, we use **ASDiv** (Miao et al., 2020) and **MAWPS** (Kadlčík et al., 2023), which fo-

cus on arithmetic word problems, as well as subject subsets from **MMLU-Pro** (Wang et al., 2024) spanning mathematics, natural sciences, social sciences, and humanities. These benchmarks differ substantially from DART-Math in both format and domain coverage.

**In-distribution evaluation.** For DART-Math, we adopt a difficulty-wise train/test split: each difficulty level is split independently, and models are trained and evaluated within the same difficulty distribution.

**Out-of-distribution (OOD) evaluation.** For OOD evaluation, POLAR is trained on the union of DART-Math training data across all difficulty levels and evaluated zero-shot. This setting directly tests whether POLAR learns transferable computation control strategies rather than heuristics specific to a dataset or difficulty level.

**Metric.** We report **pass@$k$ accuracy**, defined as the probability that at least one of the top-$k$ candidates produces a correct answer. For POLAR, the $k$ candidates correspond to the top-$k$ predicted execution programs selected via beam search. For sampling-based baselines, $k$ corresponds to the number of stochastic decoding samples. Unless otherwise stated, OOD results are reported using pass@1.

**Baselines.** We compare POLAR against standard inference and representative dynamic-computation methods. **Base** ($\tau = 0$) uses greedy decoding with temperature $\tau = 0$. **Base (sampling)** samples $k$ outputs using stochastic decoding with $\tau \in \{0.3, 0.7, 1.0\}$ and reports the best result across temperatures, increasing output diversity without altering internal execution. **DR.LLM** (Heakl et al., 2025) learns layer-routing policies from execution paths and applies them at inference time. **ShortGPT** (Men et al., 2025) statically prunes layers based on estimated importance, yielding a reduced-depth model. **MindSkip** (He et al., 2024) and **FlexiDepth** (Luo et al., 2025) learn router-based dynamic-depth policies, primarily optimized for inference efficiency. Several approaches, such as Mixture-of-Depths (Raposo et al., 2024), LaCo (Yang et al., 2024), and Mixture-of-Recursions (Bae et al., 2025), require substantial additional training or architectural modification. In contrast, POLAR performs lightweight test-time program selection without modifying pretrained model parameters.

More dataset and training details are in Appendix D.

### 4.2. Main Results

We evaluate in-distribution performance on DART-Math. Table 2 reports pass@$k$ results using *LLaMA-3.2-3B-Instruct*, with complete results provided in Appendix C.1.

**Accuracy gains arise from improved latent execution within the frozen model.** At pass@1, POLAR consistently outperforms Base (sampling) across all difficulty levels. For

*Table 3.* **Out-of-distribution (OOD) performance at pass@1.** We report accuracy using Qwen1.5-MoE-A2.7B-Chat.

| Method | ASDiv | MAWPS | MMLU-Pro | | | | | | | | | | | | |
| | | | Math | Phys | Chem | Law | Eng | Other | Econ | Health | Psych | Bus | Bio | Phil | Hist |
|---|---|---|---|---|---|---|---|---|---|---|---|---|---|---|---|
| Base ($\tau$=0) | 59.1 | 41.7 | 13.9 | 15.6 | 13.8 | 16.6 | 15.1 | 22.8 | 31.0 | 26.8 | 30.7 | 18.4 | 34.7 | 22.8 | 22.6 |
| ShortGPT | 2.3 | 0.6 | 3.8 | 0.4 | 4.2 | 3.3 | 3.8 | 2.6 | 2.4 | 2.2 | 2.8 | 4.9 | 2.9 | 1.8 | 4.7 |
| MindSkip | 0.0 | 0.0 | 0.9 | 1.2 | 1.0 | 1.9 | 1.3 | 1.3 | 0.6 | 0.7 | 1.3 | 0.8 | 1.7 | 0.6 | 1.8 |
| FlexiDepth | 0.0 | 0.0 | 2.1 | 3.5 | 2.5 | 4.4 | 3.3 | 2.5 | 1.7 | 3.5 | 2.0 | 2.9 | 3.8 | 3.0 | 3.7 |
| DR.LLM | 59.1 | 41.3 | 14.6 | 17.4 | 13.2 | 19.8 | 16.0 | 20.8 | 31.8 | 27.0 | 32.2 | 17.5 | 33.5 | 22.8 | 21.3 |
| POLAR | **63.8** | **46.7** | **18.5** | **20.3** | **18.3** | **20.4** | **19.9** | **26.6** | **34.6** | **29.5** | **35.3** | **20.9** | **36.9** | **25.9** | **23.5** |

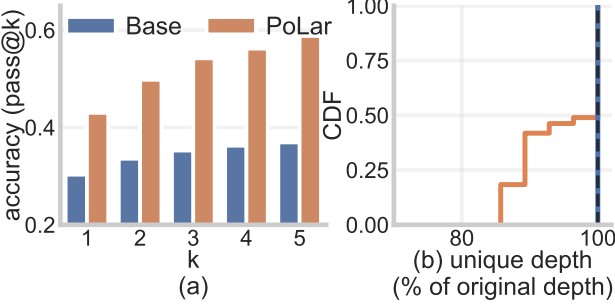

*Figure 8.* Pass@k accuracy and unique depth on Llama-3.2-3B-Instruct. (a) reports pass@$k$ accuracy for Base ($\tau = 0$) and POLAR. (b) illustrates how often POLAR generates solutions that use fewer unique layers than the original model depth.

*Table 4.* Component-wise inference overhead and end-to-end latency of POLAR on *Qwen1.5-MoE-A2.7B-Chat* with 24 layers. The additional components introduce negligible overhead compared with a standard forward pass, while POLAR reduces end-to-end latency and improves accuracy.

| Component-wise overhead | | | |
|---|---|---|---|
| **Component** | **Latency (ms)** | **Equiv. layers** | **% full forward** |
| One LLM layer | 13.23 | 1.00 | 3.5% |
| Predictor head | 0.99 | 0.07 | 0.3% |
| Beam search | 0.11 | 0.01 | 0.03% |
| Encoder | 1.95 | 0.15 | 0.5% |
| **Total additional overhead** | **3.05** | **0.23** | **0.8%** |

| End-to-end latency | | | |
|---|---|---|---|
| **Method** | **Avg. layers** | **Latency (ms)** | **Rel. / Acc. gain** |
| Base | 24.00 | 373.45 | 1.00× / − |
| POLAR (DM-1) | 23.30 | 311.41 | 0.83× / +5.8 |
| POLAR (DM-5) | 23.76 | 353.31 | 0.95× / +1.2 |

example, accuracy increases from 40.6% to 46.2% on DM-1. Since pass@1 evaluates a single decoded output, this gain reflects more effective latent execution selection rather than output-space diversity.

**Exploring the execution-program space enables effective test-time scaling.** Increasing the number of candidate execution programs ($k$) monotonically improves POLAR across all difficulty levels, evidencing strong test-time computation scaling. Figure 8(a) demonstrates monotonic pass@$k$ gains for POLAR, far exceeding the Base strategy. Crucially, Figure 8(b) shows that these gains often use fewer unique layers than a standard forward pass, indicating that improvements arise from better latent execution programs rather than increased depth. In contrast, Base (sampling) exhibits diminishing returns as $k$ increases. At pass@5, it achieves 47.6/43.2/32.8/32.8/35.6 across difficulty levels, while POLAR improves these to 68.4/48.0/46.0/40.4/45.8, yielding up to a +20.8% gain. These results show that structured execution-program exploration outperforms output sampling under a fixed computation graph.

**Program-level execution exploration is more effective than local routing decisions.** Existing dynamic-depth methods primarily make local, layer-wise routing decisions, which restrict inference to a limited execution space and often degrade accuracy in our setting. DR.LLM supports both layer skipping and repetition but operates at the

individual-layer level, limiting global coordination across depth. In contrast, POLAR formulates inference as *program-level* exploration over execution programs defined on packed contiguous segments, enabling coordinated skip and repeat patterns across depth. This design directly reflects the execution structures uncovered by MCTS, while replacing expensive search with a lightweight, learned predictor.

**POLAR incurs negligible inference overhead and reduces end-to-end latency.** Beyond counting executed layers, we measure wall-clock latency on *Qwen1.5-MoE-A2.7B-Chat* with 24 layers. As shown in Table 4, the encoder, predictor head, and beam search introduce a total additional overhead of only 3.05 ms, corresponding to 0.8% of a standard forward pass and approximately 0.23 LLM layers. This overhead is small compared with the latency reduction achieved by executing fewer layers. Consequently, POLAR reduces end-to-end latency while improving accuracy: it achieves 0.83× the base runtime on easier inputs and 0.95× on harder inputs. The learned predictor is also lightweight in parameter count: across all evaluated backbones, it contains approximately 2.1M parameters, corresponding to only 0.01%–0.06% of the base LLM size. Full parameter counts are provided in Appendix D.3.

### 4.3. Out-of-Distribution Performance

We evaluate the OOD generalization of execution programs learned from in-distribution data. As shown in Table 3, POLAR consistently outperforms standard inference on all OOD benchmarks using *Qwen1.5-MoE-A2.7B-Chat*, with full results reported in Appendix C.2.

**Execution programs learned on mathematical datasets transfer across domains.** On arithmetic word problem benchmarks such as ASDiv and MAWPS, POLAR achieves clear improvements over the standard forward pass. More notably, on MMLU-Pro, POLAR improves accuracy across diverse subject areas. We conjecture that this cross-domain transfer comes from two complementary sources. First, the external input representation maps examples from different domains into a shared semantic space, allowing the small POLAR prediction head trained on mathematics to generalize beyond its training distribution. Second, the predicted programs are constrained to simple structural patterns, namely contiguous segments with limited recurrence, which encourages reusable computation strategies rather than benchmark-specific execution heuristics.

## 5. Related Works

Transformers process inputs through sequential layer stacks, making layer-level computation reduction a critical research direction. Early-exit and layer skipping methods (Liu et al., 2020; Xin et al., 2020; Zhou et al., 2020; Liu et al., 2021a) dynamically terminate computation at intermediate layers using auxiliary classifiers and confidence metrics, allowing easy inputs to exit early. LayerSkip (Elhoushi et al., 2024) shares classifiers across layers to reduce overhead. LayerDrop (Fan et al., 2019) trains models so arbitrary layer subsets can be skipped during inference. ShortGPT (Men et al., 2025) assesses layer importance based on input-output similarity and drops low-importance layers. LaCo (Yang et al., 2024) merges layers using weight arithmetic. Recent work introduces learned routing for adaptive skipping: FlexiDepth (Luo et al., 2025) and MindSkip (He et al., 2024) attach lightweight routers to pretrained models for input-adaptive layer skipping.

In addition to skipping layers, another line of research explores layer reuse and recurrence. Universal Transformers (Dehghani et al., 2018) apply self-attention blocks recurrently with halting mechanisms to adapt depth per token. Recent looped transformers (Fan et al., 2024; Yang et al., 2023) repeatedly apply single blocks to achieve better length generalization on algorithmic tasks by adjusting loop counts during inference. The Inner Thinking Transformer (Chen et al., 2025) interleaves adaptive loops with residual "thinking" connections and per-token routing, devoting extra computation to difficult tokens. While these approaches demon-

strate the value of recurrence, they require architectural redesign and training from scratch.

Li et al. (2025) studies test-time depth adaptation by using search to dynamically skip or repeat pretrained transformer layers without finetuning. Their work demonstrates that alternative execution paths can improve inference, but the method remains search-based and requires expensive per-input program discovery. In contrast, we use MCTS only as an offline diagnostic tool to characterize the structure of the program space, and then replace search with a learned predictor that generates execution programs in a single shot.

Following this direction, DR.LLM (Heakl et al., 2025) learns routing policies from MCTS-generated supervision and supports both skipping and repeating layers. However, DR.LLM performs sequential layer-wise routing, where each decision is made locally during the forward pass and depends on intermediate hidden states. In contrast, POLAR predicts the entire execution program upfront, before executing the frozen LLM. This avoids interleaving routing with layer execution and enables more efficient inference. Moreover, DR.LLM is limited to single-layer recurrence, whereas POLAR operates on contiguous layer segments; for example, POLAR can represent multi-layer recurrent modules such as $4{\to}5{\to}4{\to}5$, which are outside the single-layer routing space. Thus, POLAR provides a more coordinated and expressive program space while preserving fully frozen base model parameters.

## 6. Conclusion

We show that inference in LLMs need not be limited to a fixed-depth forward pass. By viewing pretrained transformer layers as reusable functions, we uncover multiple valid execution programs for a single input, many of which are shorter than standard execution and can correct model errors. Motivated by this insight, we introduce POLAR, a lightweight framework that predicts input-dependent execution programs by selectively skipping or repeating contiguous layer segments at inference time, without modifying model parameters. Across models and both in-distribution and out-of-distribution benchmarks, POLAR consistently outperforms standard inference and prior dynamic-depth methods. These findings suggest that fixed-depth execution captures only a narrow subset of an LLM's latent reasoning capacity. Enabling flexible, programmatic execution over pretrained layers reallocates computation at inference time, offering a simple and effective route to more expressive and efficient inference in foundation models.

## Impact Statement

This work adapts the internal computation of pretrained LLMs at inference time by dynamically skipping or repeat-

ing layer segments. Its main potential benefit is improved efficiency: POLAR can reduce unnecessary computation on easier inputs while allocating more latent computation to harder ones, lowering inference cost, latency, and energy use without retraining the base model. This may make capable LLMs more accessible to researchers and practitioners with limited compute, and may support more sustainable deployment of foundation models. As with other methods that improve LLM capability or efficiency, broader deployment may amplify both beneficial and harmful uses. Potential benefits include education, scientific reasoning, and software assistance, while potential misuse includes scalable generation of misleading or harmful content. These risks largely arise from the underlying pretrained models and their applications rather than from dynamic execution itself. Since POLAR changes the execution path in an input-dependent manner, future work may further study how such paths can be audited or interpreted. Our experiments focus on mathematical reasoning and related benchmarks. Before applying dynamic execution in high-stakes domains, future work should evaluate robustness, calibration, interpretability, and safety alongside accuracy and efficiency. We hope this work encourages more responsible test-time computation methods that improve model performance while making compute allocation more transparent and efficient.

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

# A. Related Work

**Layer Pruning and Early-Exit Neural Networks**    Many works aim to accelerate large Transformers by statically pruning weights or dynamically halting computation. Static pruning typically removes redundant neurons, heads, or layers after training. For example, Liu et al. (2021b) demonstrate that a significant fraction of BERT's attention heads can be dropped with negligible performance loss, and Gordon et al. (2020) investigate fine-grained weight pruning in BERT. (Fan et al., 2019) introduce LayerDrop, a structured dropout technique that effectively trains models so arbitrary subsets of layers can be skipped during inference without requiring fine-tuning. These methods produce smaller models that trade computation for a small accuracy loss.

By contrast, early-exit or input-adaptive methods add auxiliary classifiers at intermediate layers so that "easy" inputs exit early. Notable examples include FastBERT (Liu et al., 2020) and DeeBERT (Xin et al., 2020), which insert classifiers after each block and use confidence or entropy metrics to decide when to stop. PABEE (Zhou et al., 2020) employs a patience criterion to halt when predictions stabilize. DACT-BERT (Eyzaguirre et al., 2022) adopts a differentiable Adaptive Computation Time mechanism to learn how many Transformer layers to run for each example. Liu et al. (2021a) estimate input "hardness" via mutual information or reconstruction error to pre-determine the number of Transformer layers to use.

These early-exit networks achieve significant speedups on NLP tasks by adaptively reducing depth per input. More recently, early-exit ideas have been extended to vision and multimodal Transformers. Xu et al. (2023) propose LGViT, which adds heterogeneous exit heads (local and global) to ViT so that vision transformers can terminate early with minimal feature loss. Tang et al. (2023) introduce MuE ("Multiple Exiting"), a strategy for unified vision-language models that dynamically skips layers in both encoder and decoder based on input similarity. These works demonstrate that later layers can be skipped to allow image and vision-language models to adapt computation per sample with minimal accuracy drop. Our work generalizes this approach by allowing skipping of arbitrary layers and enabling reuse of certain layers.

**Looped Transformer and Recurrent Depth**    Another line of research makes Transformer depth adaptive by looping or repeating layers. The Universal Transformer (Dehghani et al., 2018) was an early example: it applies the same self-attention block recurrently and uses a halting mechanism to determine when each position is "done" (adapting depth per token). Building on these ideas, recent work explicitly introduces loops in model architectures. Fan et al. (2024) demonstrate that a Looped Transformer – a single Transformer block applied repeatedly – can achieve much better length generalization on algorithmic tasks by adjusting the number of loops during inference. Similarly, Yang et al. (2023) note that looped architectures excel at learning algorithms by explicitly incorporating iterative characteristics into the transformer architecture. More sophisticated variants like the Inner Thinking Transformer Chen et al. (2025) interleave adaptive loops with residual "thinking" connections and per-token routing, enabling the model to devote extra computation only to particularly difficult tokens. In summary, these approaches explore recurrent or elastic depth via explicit loops to tailor the number of applied layers to each input's complexity. Unlike our approach, they require special architecture design and training from scratch, whereas our work focuses on pure test-time adaptation.

**Dynamic Routing and Modular Inference**    A third theme treats networks as collections of modules or experts with dynamically chosen pathways per sample. Mixture-of-Experts (MoE) Transformer layers are a well-known example: they maintain multiple sub-networks ("experts") and route each token to a subset. Wu et al. (2024) introduce Routing Experts (RoE) for multimodal LLMs, retrofitting trained models into a mixture-of-experts style by learning input-dependent shortcut routes through layers, guided by sparsity regularizers. Jain et al. (2024) present Mixture of Nested Experts (MoNE): experts organized in a hierarchy of increasing capacity, where tokens are sent to smaller experts when sufficient. MoNE learns to prioritize easy tokens through low-cost experts and reserve full models for hard cases, halving inference compute on ImageNet/Video tasks.

These methods exemplify sample-wise routing: at inference time, the model conditionally activates different sub-modules or experts for each input. Similarly, neural module networks (Andreas et al., 2016) assemble task-specific computation graphs from a library of modules. In modern LLMs/VLMs, these routing approaches – whether through gating experts, skipping layers, or assembling modules – form a spectrum of modular inference techniques that adapt the computation graph on a per-sample basis to balance cost and accuracy. Interestingly, our work suggests that transformer layers can function effectively as modules even without being specifically trained for that purpose.

# B. Searching the Execution Program Space

This appendix provides full details of the execution-program search procedure used to test the conjecture in Section 2. The search is used purely as a diagnostic tool to study the existence and structure of valid execution programs, rather than as a practical inference-time method.

## B.1. Execution Program Space

We follow the formalization in the main text and represent inference as the execution of a variable-length program that composes pretrained transformer layers. Consider a pretrained LLM with $D$ transformer layers, where each layer defines a fixed computation function

$$f_i : \mathbb{R}^{T \times d} \to \mathbb{R}^{T \times d}, \quad i \in \{0, \ldots, D-1\}.$$

An execution program is defined as a finite sequence of layer indices

$$\pi = (i_1, i_2, \ldots, i_K), \quad i_k \in \{0, \ldots, D-1\},$$

which induces the composed computation

$$F_\pi = f_{i_K} \circ \cdots \circ f_{i_1}.$$

Executing a program applies this composition to the input representation and produces a prediction. A program is considered *valid* for a given input if it yields a correct prediction.

Programs may be shorter than the standard forward pass through layer skipping, or longer through layer repetition. Increasing program length corresponds to increasing the number of latent reasoning steps.

## B.2. Search Space Constraints

The unconstrained space of programs grows exponentially with program length. To make search tractable while preserving expressiveness, we restrict the action space to structured operations on contiguous subsequences of layer indices. Specifically, we allow two classes of actions:

- **Skip**: remove a contiguous block of $k$ indices from the program;

- **Repeat**: duplicate a contiguous block of $k$ indices for $r$ repetitions.

In all experiments, block size $k$ and repetition count $r$ are bounded by small constants ($k, r \leq 4$). These constraints significantly reduce the branching factor while retaining the ability to realize layer skipping, recurrence, and emergent reordering patterns.

## B.3. Monte Carlo Tree Search Formulation

We formulate program discovery as a sequential decision process and employ Monte Carlo Tree Search (MCTS) to explore the constrained program space.

**State and Actions.** Each MCTS node corresponds to a partial or complete execution program $\pi$. Actions modify the current program by applying a valid skip or repeat operation, yielding a new program.

**Reward.** For a completed program $\pi$ and input $x$ with ground-truth answer $y$, we define a binary reward

$$r(\pi, x) = \mathbf{1}\{F_\pi(x) = y\},$$

where $F_\pi(x)$ denotes executing the composed computation induced by $\pi$.

**Tree Policy.** Tree traversal is guided by a UCB-style objective that balances exploitation, exploration, and program length regularization:

$$\text{UCB}(\pi) = \frac{R(\pi)}{v(\pi)} + c\sqrt{\frac{\ln V}{v(\pi)}} - \lambda\frac{|\pi|}{D},$$

where $R(\pi)$ is the cumulative reward, $v(\pi)$ is the visit count, $V$ is the total number of simulations, and $\lambda$ penalizes long programs to encourage efficiency.

---

**Algorithm 1** Monte Carlo Tree Search for Execution Programs

---

**Require:** Input $x$, number of simulations $N_{\text{sim}}$
 1: Initialize root program $\pi_0 = (0, 1, \ldots, D - 1)$
 2: **for** $n = 1$ to $N_{\text{sim}}$ **do**
 3:     **Selection:** traverse tree using UCB to reach a leaf program
 4:     **Expansion:** generate child programs via valid skip/repeat actions
 5:     **Simulation:** execute $F_\pi(x)$ and compute reward $r(\pi, x)$
 6:     **Backpropagation:** update visit counts and cumulative rewards
 7: **end for**
 8: **return** explored programs and associated statistics

---

### B.4. Search Algorithm

Algorithm 1 summarizes the MCTS procedure. We initialize the root node with the standard forward execution $\pi_0 = (0, 1, \ldots, D - 1)$ and iteratively perform selection, expansion, simulation, and backpropagation. After a fixed number of simulations, we collect all explored programs with nonzero visit counts and analyze their validity and structural properties.

# C. Experimental Results

This appendix provides additional experimental results that complement the main paper. We report full quantitative comparisons for both in-distribution and out-of-distribution evaluations across multiple pretrained LLMs. Unless otherwise stated, all models are evaluated in a fully frozen setting, and POLAR only predicts execution programs at inference time.

## C.1. In-Distribution Performance

We first report detailed in-distribution results on DART-Math, a structured mathematical reasoning benchmark with five difficulty levels (DM-1 to DM-5). Tables 5, 6, and 7 present pass@k accuracy under different inference strategies for Qwen1.5-MoE-A2.7B-Chat, Qwen2.5-3B-Instruct, and Qwen3-8B, respectively.

Across all models and difficulty levels, POLAR consistently outperforms standard inference and prior dynamic-depth baselines. In particular, increasing $p@k$ leads to monotonic accuracy improvements for POLAR, demonstrating effective test-time computation scaling through execution-program exploration. At $p@5$, POLAR achieves substantial absolute gains over Base (sampling), with improvements of +22.0 / +18.4 / +14.4 / +10.4 / +11.4 on Qwen1.5-MoE-A2.7B-Chat, +17.6 / +10.4 / +7.8 / +2.2 / +9.8 on Qwen2.5-3B-Instruct, and +7.2 / +17.0 / +11.8 / +5.4 / +7.2 on Qwen3-8B for DM-1 to DM-5, respectively.

Notably, methods that rely solely on layer skipping (e.g., ShortGPT, MindSkip, FlexiDepth) often suffer severe accuracy degradation, especially on harder difficulty levels. In contrast, POLAR jointly supports layer skipping and recurrence, allowing it to retain or improve accuracy while exploring diverse execution programs. Compared to DR.LLM, which performs layer-level routing, POLAR consistently achieves higher accuracy, particularly at larger $p@k$, indicating the benefit of structured, program-level execution prediction.

## C.2. Out-of-Distribution Generalization

We further evaluate the out-of-distribution (OOD) generalization of POLAR on benchmarks that differ substantially from DART-Math in both format and domain. Tables 8, 9, and 10 report pass@1 accuracy on ASDiv, MAWPS, and subject-wise subsets of MMLU-Pro using LLaMA-3.2-3B-Instruct, Qwen2.5-3B-Instruct, and Qwen3-8B, respectively.

Across all evaluated models, POLAR consistently improves over standard inference on arithmetic word problem benchmarks (ASDiv and MAWPS), indicating strong transfer from structured mathematical reasoning to natural language problem settings. On MMLU-Pro, which spans diverse domains including mathematics, natural sciences, social sciences, and humanities, POLAR achieves broad and consistent gains across most subject areas.

These results suggest that the execution programs learned by POLAR capture general, transferable computation control strategies rather than dataset-specific heuristics. Despite being trained on mathematical reasoning data, POLAR generalizes effectively to heterogeneous domains, highlighting the robustness of program-of-layers inference and its applicability beyond the original training distribution.

*Table 5.* **Pass@k accuracy under different inference strategies applied to Qwen1.5-MoE-A2.7B-Chat.** For difficulty levels DM-1 to DM-5, POLAR consistently outperforming the Base program across all $k$ values, where $\tau = 0$ denotes zero temperature.

| Method | p@ | DM-1 | DM-2 | DM-3 | DM-4 | DM-5 |
|---|---|---|---|---|---|---|
| **Qwen1.5-MoE-A2.7B-Chat** | | | | | | |
| Base ($\tau$=0) | 1 | 38.0 | 24.6 | 20.6 | 15.2 | 14.2 |
| | 1 | 35.4 | 22.0 | 14.6 | 12.6 | 6.6 |
| | 2 | 36.4 | 24.2 | 15.8 | 13.6 | 8.2 |
| Base (sampling) | 3 | 38.8 | 24.6 | 16.6 | 14.0 | 9.6 |
| | 4 | 39.6 | 25.0 | 17.8 | 14.8 | 11.2 |
| | 5 | 40.0 | 25.6 | 18.6 | 15.0 | 11.8 |
| | 1 | 19.6 | 12.2 | 7.2 | 5.4 | 3.4 |
| | 2 | 22.4 | 15.0 | 9.4 | 7.2 | 5.2 |
| ShortGPT | 3 | 25.6 | 16.6 | 10.6 | 8.6 | 6.2 |
| | 4 | 26.6 | 17.6 | 11.4 | 9.6 | 7.0 |
| | 5 | 27.2 | 18.6 | 12.4 | 10.2 | 7.6 |
| | 1 | 0.0 | 0.0 | 0.0 | 0.0 | 0.0 |
| | 2 | 0.0 | 0.0 | 0.0 | 0.0 | 0.0 |
| MindSkip | 3 | 0.0 | 0.0 | 0.0 | 0.0 | 0.0 |
| | 4 | 0.0 | 0.0 | 0.0 | 0.0 | 0.0 |
| | 5 | 0.0 | 0.0 | 0.0 | 0.0 | 0.0 |
| | 1 | 0.0 | 0.0 | 0.0 | 0.0 | 0.0 |
| | 2 | 0.0 | 0.4 | 0.2 | 0.0 | 0.0 |
| FlexiDepth | 3 | 0.0 | 0.8 | 0.2 | 0.0 | 0.0 |
| | 4 | 0.4 | 0.8 | 0.2 | 0.0 | 0.0 |
| | 5 | 0.4 | 1.0 | 0.2 | 0.0 | 0.0 |
| | 1 | 38.0 | 24.6 | 20.6 | 15.2 | 14.2 |
| | 2 | 43.8 | 29.8 | 23.2 | 18.2 | 18.0 |
| DR.LLM | 3 | 44.4 | 30.2 | 23.4 | 18.2 | 18.8 |
| | 4 | 44.4 | 30.4 | 23.8 | 18.2 | 20.0 |
| | 5 | 48.4 | 36.8 | 27.4 | 22.8 | 21.6 |
| | 1 | 43.8 | 26.2 | 22.0 | 17.2 | 15.4 |
| | 2 | 53.2 | 34.6 | 28.2 | 20.8 | 19.8 |
| **POLAR** | 3 | 58.0 | 39.0 | 29.0 | 22.4 | 21.6 |
| | 4 | 60.8 | 41.2 | 31.2 | 23.6 | 22.6 |
| | 5 | **62.0** | **44.0** | **33.0** | **25.4** | **23.2** |
| $\Delta$ vs. Base (sampling) | 5 | **+22.0** | **+18.4** | **+14.4** | **+10.4** | **+11.4** |

*Table 6.* **Pass@k accuracy under different inference strategies applied to Qwen2.5-3B-Instruct.** For difficulty levels DM-1 to DM-5, POLAR consistently outperforming the Base program across all $k$ values, where $\tau = 0$ denotes zero temperature.

| Method | p@ | DM-1 | DM-2 | DM-3 | DM-4 | DM-5 |
|---|---|---|---|---|---|---|
| **Qwen2.5-3B-Instruct** | | | | | | |
| Base ($\tau$=0) | 1 | 22.0 | 13.6 | 11.0 | 7.8 | 5.2 |
| | 1 | 24.2 | 16.0 | 11.6 | 8.2 | 5.4 |
| | 2 | 32.6 | 22.8 | 16.0 | 12.2 | 8.6 |
| Base (sampling) | 3 | 37.8 | 26.8 | 18.2 | 13.8 | 10.0 |
| | 4 | 40.6 | 29.2 | 19.2 | 15.2 | 10.6 |
| | 5 | 42.2 | 30.2 | 20.4 | 15.8 | 13.0 |
| | 1 | 0.4 | 0.0 | 0.0 | 0.0 | 0.0 |
| | 2 | 0.8 | 0.0 | 0.0 | 0.0 | 0.0 |
| ShortGPT | 3 | 1.0 | 0.0 | 0.0 | 0.0 | 0.0 |
| | 4 | 1.2 | 0.0 | 0.0 | 0.0 | 0.0 |
| | 5 | 1.4 | 0.0 | 0.0 | 0.0 | 0.0 |
| | 1 | 0.6 | 0.4 | 0.0 | 0.4 | 0.0 |
| | 2 | 1.4 | 0.6 | 0.0 | 0.4 | 0.0 |
| MindSkip | 3 | 1.4 | 0.8 | 0.0 | 0.4 | 0.0 |
| | 4 | 1.8 | 1.0 | 0.0 | 0.4 | 0.0 |
| | 5 | 1.8 | 1.4 | 0.0 | 0.4 | 0.0 |
| | 1 | 30.0 | 8.8 | 3.2 | 1.0 | 1.2 |
| | 2 | 43.6 | 15.0 | 6.4 | 2.2 | 1.8 |
| FlexiDepth | 3 | 49.6 | 20.0 | 8.6 | 2.8 | 3.0 |
| | 4 | 55.4 | 22.4 | 11.0 | 3.4 | 4.6 |
| | 5 | 60.0 | 24.0 | 12.0 | 3.8 | 5.0 |
| | 1 | 6.8 | 6.6 | 4.6 | 2.2 | 4.2 |
| | 2 | 14.2 | 9.8 | 7.8 | 3.6 | 7.2 |
| DR.LLM | 3 | 17.4 | 11.8 | 9.6 | 5.4 | 7.4 |
| | 4 | 20.8 | 13.0 | 11.2 | 7.0 | 9.4 |
| | 5 | 21.6 | 14.6 | 12.6 | 8.0 | 10.2 |
| | 1 | 44.4 | 24.4 | 20.2 | 11.8 | 12.8 |
| | 2 | 50.6 | 28.0 | 22.2 | 12.8 | 16.4 |
| **POLAR** | 3 | 54.2 | 31.4 | 25.0 | 14.8 | 19.0 |
| | 4 | 57.4 | 34.6 | 26.8 | 17.2 | 21.0 |
| | 5 | **59.8** | **40.6** | **28.2** | **18.0** | **22.8** |
| $\Delta$ vs. Base (sampling) | 5 | **+17.6** | **+10.4** | **+7.8** | **+2.2** | **+9.8** |

*Table 7.* **Pass@k accuracy under different inference strategies applied to Qwen3-8B.** For difficulty levels DM-1 to DM-5, POLAR consistently outperforming the Base program across all $k$ values, where $\tau = 0$ denotes zero temperature.

| Method | p@ | DM-1 | DM-2 | DM-3 | DM-4 | DM-5 |
|---|---|---|---|---|---|---|
| **Qwen3-8B** | | | | | | |
| Base ($\tau$=0) | 1 | 41.6 | 28.8 | 19.0 | 12.0 | 13.0 |
| Base (sampling) | 1 | 37.0 | 23.0 | 9.2 | 9.6 | 7.0 |
| | 2 | 42.0 | 25.6 | 11.6 | 10.6 | 8.8 |
| | 3 | 45.8 | 26.0 | 12.4 | 11.4 | 9.8 |
| | 4 | 47.0 | 26.6 | 12.6 | 12.8 | 10.2 |
| | 5 | 48.4 | 27.8 | 13.6 | 13.4 | 10.6 |
| ShortGPT | 1 | 0.0 | 0.0 | 0.0 | 0.0 | 0.0 |
| | 2 | 0.0 | 0.0 | 0.0 | 0.0 | 0.0 |
| | 3 | 0.0 | 0.0 | 0.0 | 0.0 | 0.0 |
| | 4 | 0.0 | 0.0 | 0.0 | 0.0 | 0.0 |
| | 5 | 0.0 | 0.0 | 0.0 | 0.0 | 0.0 |
| MindSkip | 1 | 0.0 | 0.0 | 0.0 | 0.0 | 0.0 |
| | 2 | 0.0 | 0.0 | 0.0 | 0.0 | 0.0 |
| | 3 | 0.0 | 0.0 | 0.0 | 0.0 | 0.0 |
| | 4 | 0.0 | 0.0 | 0.0 | 0.0 | 0.0 |
| | 5 | 0.0 | 0.0 | 0.0 | 0.0 | 0.0 |
| FlexiDepth | 1 | 49.6 | 29.4 | 8.8 | 1.8 | 1.0 |
| | 2 | 67.0 | 42.2 | 13.8 | 3.2 | 2.0 |
| | 3 | 75.8 | 50.6 | 16.4 | 4.2 | 2.8 |
| | 4 | 81.2 | 55.6 | 18.4 | 5.2 | 4.8 |
| | 5 | 84.8 | 59.2 | 19.4 | 6.8 | 6.6 |
| DR.LLM | 1 | 2.0 | 0.6 | 0.6 | 0.2 | 2.0 |
| | 2 | 2.4 | 0.6 | 0.8 | 0.2 | 2.4 |
| | 3 | 2.4 | 0.6 | 0.8 | 0.2 | 2.4 |
| | 4 | 2.4 | 0.6 | 0.8 | 0.2 | 2.8 |
| | 5 | 2.4 | 1.0 | 0.8 | 0.2 | 2.8 |
| **POLAR** | 1 | 43.2 | 30.4 | 20.8 | 13.5 | 14.9 |
| | 2 | 48.4 | 33.2 | 21.2 | 16.0 | 14.6 |
| | 3 | 50.8 | 41.0 | 22.2 | 18.0 | 17.2 |
| | 4 | 53.4 | 44.0 | 24.8 | 18.4 | 17.6 |
| | 5 | **55.6** | **44.8** | **25.4** | **18.8** | **17.8** |
| $\Delta$ vs. Base (sampling) | 5 | **+7.2** | **+17.0** | **+11.8** | **+5.4** | **+7.2** |

*Table 8.* **Out-of-distribution (OOD) performance at pass@1.** We report accuracy on OOD benchmarks using LLaMA-3.2-3B-Instruct.

| Method | ASDiv | MAWPS | MMLU-Pro | | | | | | | | | | | | |
|---|---|---|---|---|---|---|---|---|---|---|---|---|---|---|---|
| | | | Math | Phys | Chem | Law | Eng | Other | Econ | Health | Psych | Bus | Bio | Phil | Hist |
| Base ($\tau$=0) | 78.4 | 71.5 | 19.5 | 19.0 | 18.6 | 17.1 | 19.8 | 22.4 | 36.1 | 28.5 | 33.0 | 22.2 | 44.9 | 23.8 | 28.3 |
| ShortGPT | 2.3 | 4.6 | 29.9 | 26.9 | 26.9 | 23.3 | 20.1 | 33.0 | 36.7 | 35.3 | 38.2 | 28.6 | 36.0 | 41.7 | 40.9 |
| MindSkip | 9.0 | 7.5 | **40.4** | **47.7** | **43.3** | 31.8 | **43.7** | 38.5 | 47.6 | 47.2 | 51.5 | 31.7 | 52.3 | **49.7** | **55.6** |
| FlexiDepth | 4.7 | 1.3 | 7.8 | 5.9 | 4.8 | 6.4 | 6.3 | 4.7 | 5.9 | 5.9 | 3.6 | 6.0 | 4.9 | 7.4 | 4.7 |
| DR.LLM | 75.4 | 61.3 | 13.6 | 14.4 | 16.2 | 10.0 | 16.4 | 9.0 | 13.6 | 10.6 | 15.2 | 13.0 | 19.8 | 8.2 | 23.1 |
| POLAR | **81.4** | **73.7** | 40.1 | 39.0 | 41.0 | **34.3** | 43.2 | **42.9** | **50.1** | **52.4** | 52.4 | **32.7** | **56.6** | 44.5 | 48.8 |

*Table 9.* **Out-of-distribution (OOD) performance at pass@1.** We report accuracy on OOD benchmarks using Qwen2.5-3B-Instruct.

| Method | ASDiv | MAWPS | MMLU-Pro | | | | | | | | | | | | |
| --- | --- | --- | --- | --- | --- | --- | --- | --- | --- | --- | --- | --- | --- | --- | --- |
| | | | Math | Phys | Chem | Law | Eng | Other | Econ | Health | Psych | Bus | Bio | Phil | Hist |
| Base ($\tau$=0) | 49.5 | 36.2 | 26.3 | 28.7 | 27.7 | 24.4 | 37.0 | 33.9 | 47.5 | 40.3 | 52.6 | 31.3 | 62.2 | 33.7 | 34.6 |
| ShortGPT | 1.3 | 1.2 | 8.9 | 9.3 | 10.0 | 12.1 | 12.1 | 12.8 | 12.6 | 13.4 | 8.9 | 11.5 | 10.3 | 9.8 | 12.1 |
| MindSkip | 8.0 | 3.5 | 7.6 | 7.5 | 8.3 | 6.0 | 9.5 | 7.4 | 8.6 | 9.8 | 7.5 | 8.5 | 7.5 | 9.4 | 6.8 |
| FlexiDepth | 2.7 | 0.6 | 9.8 | 9.3 | 10.5 | 10.4 | 11.4 | 11.5 | 10.8 | 11.5 | 9.9 | 8.2 | 10.6 | 11.4 | 11.0 |
| DR.LLM | 0.0 | 0.0 | 4.0 | 4.8 | 2.8 | 3.2 | 3.0 | 3.2 | 2.8 | 2.6 | 1.4 | 4.6 | 5.8 | 3.0 | 4.2 |
| POLAR | **78.1** | **57.7** | **32.1** | **35.6** | **33.0** | **27.6** | **41.5** | **38.9** | **53.0** | **46.5** | **57.4** | **33.4** | **66.0** | **38.9** | **35.4** |

*Table 10.* **Out-of-distribution (OOD) performance at pass@1.** We report accuracy on OOD benchmarks using Qwen3-8B.

| Method | ASDiv | MAWPS | MMLU-Pro | | | | | | | | | | | | |
| --- | --- | --- | --- | --- | --- | --- | --- | --- | --- | --- | --- | --- | --- | --- | --- |
| | | | Math | Phys | Chem | Law | Eng | Other | Econ | Health | Psych | Bus | Bio | Phil | Hist |
| Base ($\tau$=0) | 67.1 | 52.1 | 21.4 | 24.8 | 19.2 | 32.6 | 23.4 | 43.0 | 60.4 | 56.2 | 62.8 | 26.6 | 72.2 | 42.3 | 50.1 |
| ShortGPT | 0.0 | 0.0 | 0.0 | 0.0 | 0.0 | 0.0 | 0.0 | 0.0 | 0.0 | 0.0 | 0.0 | 0.0 | 0.0 | 0.0 | 0.0 |
| MindSkip | 20.9 | 13.1 | 26.1 | 33.9 | 31.5 | **58.0** | 32.1 | **67.0** | 64.7 | **68.0** | 71.9 | **36.6** | 68.8 | **68.5** | **65.9** |
| FlexiDepth | 1.0 | 0.4 | 0.9 | 1.0 | 0.6 | 0.6 | 0.5 | 0.8 | 0.9 | 1.0 | 0.1 | 0.8 | 1.0 | 0.6 | 1.3 |
| DR.LLM | 69.4 | 0.0 | 0.6 | 0.2 | 1.8 | 1.8 | 0.8 | 0.6 | 1.2 | 1.4 | 2.0 | 1.2 | 0.8 | 0.6 | 0.8 |
| POLAR | **72.8** | **66.9** | **26.8** | **34.6** | **23.4** | 48.7 | **35.8** | 57.1 | **65.2** | 67.4 | **72.4** | 34.1 | **75.4** | 56.7 | 62.0 |

# D. Empirical Details

## D.1. Dataset Details

All in-distribution experiments are conducted on **DART-Math**, a structured mathematical reasoning benchmark consisting of five difficulty levels, denoted as **DM-1** to **DM-5**.

Each difficulty level contains 2,000 problem instances, resulting in 10,000 examples in total across all difficulty levels. We adopt a difficulty-wise data split, where each difficulty level is split independently into training, validation, and test sets. Specifically, for each difficulty level, we use:

- 1,250 examples for training,

- 250 examples for validation, and

- 500 examples for testing.

Overall, this results in 6,250 training examples, 1,250 validation examples, and 2,500 test examples. All in-distribution results are reported on the held-out test sets corresponding to the same difficulty level used for training.

## D.2. Training Configuration

We train the POLAR prediction network using supervised learning on the training splits described above. All hyperparameters are selected via validation tuning.

**Optimization.** We use the AdamW optimizer and tune the learning rate from {1e-4, 3e-4, 5e-4, 8e-4, 1e-3, 3e-3}, the batch size from {32, 128, 256}, and the number of training epochs from {3, 10} based on validation performance. We adopt a cosine learning rate schedule with a linear warmup of `warmup_steps` = 10 steps. Unless otherwise specified, the best-performing configuration on the validation set is used for reporting results.

**Output length.** The maximum number of generated output tokens is set to **50**, consistent with the output length used when collecting supervision via MCTS, as the model is instructed to directly generate the final answer rather than intermediate reasoning steps.

## D.3. Predictor Size

The learned POLAR predictor is lightweight compared with the frozen base LLM. Across the evaluated models, the predictor contains approximately 2.1M parameters, corresponding to only 0.01%–0.06% of the base model size. This small footprint makes training and inference inexpensive relative to standard LLM fine-tuning or full-model execution.

*Table 11.* Parameter size of the learned POLAR predictor compared with each frozen base LLM.

| Model | Base LLM params | Predictor params | Predictor / Base |
|---|---|---|---|
| Qwen1.5-MoE-A2.7B-Chat | 14.32B | 2.11M | 0.0148% |
| Qwen2.5-3B-Instruct | 3.40B | 2.12M | 0.0623% |
| Qwen3-8B | 8.19B | 2.12M | 0.0258% |
| LLaMA-3.2-3B-Instruct | 3.61B | 2.11M | 0.0586% |

## D.4. Direct Prompting

We adopt a direct prompting strategy throughout all experiments, without eliciting chain-of-thought or intermediate reasoning. The model is explicitly instructed to output only the final answer in a strictly formatted form.

Given a math problem instance `question`, the input prompt is constructed as follows:

```
Solve the following math problem and output ONLY the final answer directly,
formatted strictly as \boxed{ANSWER}.
```

```
### Problem Start
{question}
### Problem End
Answer:
```

This prompt design enforces concise answer generation and isolates the effect of latent execution programs from token-level reasoning strategies.

