# OpenReview forum: "Skip a Layer or Loop It? Learning Program-of-Layers in LLMs"
_ICML.cc/2026/Conference — ICML 2026 spotlight_

### Official Review · Reviewer_CVa6 · 2026-02-26

**Soundness:** 4
**Presentation:** 3
**Significance:** 3
**Originality:** 3
**Overall Recommendation:** 5
**Confidence:** 4

**Summary:**

The paper studies the feasibility of deciding whether to skip or loop transformer layers at inference time without fine-tuning the model or changing its architecture. The authors first analyze the behavior of models with selectively looped or removed layers when the decisions to loop or skip are made through a Monte Carlo tree search over layer actions (skip, loop, keep as-is). They find that modifying transformers with such computations improves performance and often even results in shorter “programs” that execute tasks. Then, the authors design a lightweight module (a separate transformer model) that determines these layer actions without the sequential overhead of the Monte Carlo search. They evaluate this approach and confirm that it, too, improves model performance (including OOD generalization), often results in shorter execution paths, and can even correct original model mistakes.

**Compliance With Llm Reviewing Policy:**

Affirmed.

**Final Justification:**

The paper presents an innovative study of how computation depth can be controlled in transformers. It is well written and can help guide further work on similar models, so I believe it benefits the community.

**Key Questions For Authors:**

1. I believe this wasn’t discussed in the paper: What are the runtime overheads of
	1. the MCTS action selection, and
	2. the learned module
for the total runtime?
2. On a related note, how large is the trained learned module (parameter counts)? I could not find that in the draft.
3. To my understanding, the learned module only allows for a single repetition of layer modules. What if the training data (the MCTS trace) requires more than one loop? Are such training instances discarded?

**Limitations:**

yes

**Strengths And Weaknesses:**

- Presentation: A bit of a nit, but I found the phrasing of Lines 066–078 in the Introduction slightly confusing, as they are written in terms of aims rather than contributions after already discussing some contributions of the paper. Thus, it wasn’t clear to me whether this is describing contributions or future work directions.
	- Relatedly, I feel like the draft could, at least for my first reading, benefit from a short paragraph at the end of the Introduction outlining the structure of the paper. That’s mainly because the paper, in a sens,e has two parts (the MCTS and the lightweight module part), so outlining that in advance could be useful.
	- Aside from that, I found the motivation and the exposition of the paper (both the methodology and the results) clear and easy to follow.
- Significance: The paper tackles the active research area of transformers with a dynamic allocation of resources (e.g., looped transformers), where much is still unknown. With an original idea of how to allocate resources, I think the paper makes a meaningful and interesting contribution to the field.
- Originality: I believe the use of MCTS for per-layer decisions is novel and very viable. The method is evaluated thoroughly, found to be beneficial, and then translated to a more practical mechanism, which is motivated and explained well.
- Originality: In particular, I liked the analysis of the optimal MCTS traces and the simplifications made based on it (looping at most once, grouping layers into modules, etc.). This led to the design of a useful and tractable simplification.
- Soundness: The experiments are thorough, and the baselines are, to the best of my knowledge, appropriate and well-evaluated.
	- Relatedly, related work is covered well and provides a useful lens onto the contributions of this paper.

---

> ### Author Rebuttal · Authors · 2026-03-31
>
> We thank the reviewer for the very positive and thoughtful evaluation. We are especially encouraged that the reviewer recognizes the value of our analysis of MCTS traces and the resulting simplifications (e.g., short segments and limited recurrence), as well as the practical formulation via a learned prediction module.
>
> We address the questions and suggestions below.
>
> ---
>
> > Q1. **Presentation: Lines 066–078 are slightly confusing … structure of the paper could be clearer**
>
> We thank the reviewer for this helpful suggestion. We will revise the Introduction to clearly distinguish between contributions and motivations, and add a short paragraph for the paper outline, i.e., MCTS analysis → design of the learned module → experiments, to improve readability.
>
> ---
>
> > Q2. **What are the runtime overheads of (a) MCTS action selection and (b) the learned module?**
>
> We clarify the roles and costs of each component:
>
> **(a) MCTS.**
> MCTS is used in *only offline training* for data collection and is **not part of inference-time computation**. It requires access to ground-truth answers and performs multiple forward passes per input, determined by the number of simulations (100 in our experiments). This corresponds to ~100× the cost of a standard forward pass, making it orders of magnitude more expensive. This cost is incurred once during training data curation and does not affect deployment.
>
> **(b) Learned module (POLAR).**
> At inference time, POLAR replaces MCTS with a lightweight predictor and a small beam search, whose total overhead is 3.05 ms **(0.8% of a full forward pass, <1 LLM layer, Table X)**, which is negligible compared to a standard forward pass (~373 ms). This overhead is associated with great reduction in execution depth (fewer layers applied). So POLAR often leads to lower end-to-end latency **(0.83× for easier inputs and 0.95× for harder inputs, Table Y)**.
>
> **In summary:** MCTS incurs ~100× the cost of a forward pass (offline), while POLAR adds only **0.8% overhead** at inference and often results in **lower overall runtime (0.83×–0.95× of a standard forward pass)**.
>
> We will include the following detailed analysis in the revision:
>
> **Table X. Component-wise inference overhead of POLAR.** (Qwen1.5-MoE-A2.7B-Chat, 24 layers)
> |Component|Latency (ms)|Equivalent LLM layers|% of full forward|
> |-|-:|-:|-:|
> |1 LLM layer|13.23|1.0|3.5%|
> |Predictor head|0.99|0.1|0.3%|
> |Beam search|0.11|0.01|0.03%|
> |**Predictor + beam**|**1.10**|**0.1**|**0.3%**|
> |Encoder (amortized)|1.95|0.1|0.5%|
> |**Total overhead**|**3.05**|**0.2**|**0.8%**|
>
> **Table Y. End-to-end latency, execution depth, and accuracy.**
> |Method|Avg. layers|Latency (ms)|Rel. to Base|Accuracy (Δ vs Base)|
> |-|-:|-:|-:|-:|
> |Base (24 layers)|24.00|373.45|1.00x| — |
> |POLAR (DM-1)|23.30|311.41|0.83x|**+5.8**|
> |POLAR (DM-5)|23.76|353.31|0.95x|**+1.2**|
>
> ---
>
> > Q3. **How large is the learned module (parameter counts)?**
>
> The learned module is extremely lightweight compared to the base LLM. Across all models, it contains only ~2.1M parameters, which corresponds to **0.01%–0.06% of the base model size** (e.g., 2.11M vs 14.32B for Qwen1.5-MoE-A2.7B).
>
> This small footprint is consistent across different model scales (3B–14B), making the predictor negligible in both memory and compute relative to the base LLM.
>
> **Table Z. Learned Module Parameter Sizes**
> |Model|Base LLM|Learned pred.|Learned predictor params / Base LLM params|
> |-|-|-|-|
> |Qwen1.5-MoE-A2.7B-Chat|14.32B|2.11M|0.0148%|
> |Qwen2.5-3B-Instruct|3.40B|2.12M|0.0623%|
> |Qwen3-8B|8.19B|2.12M|0.0258%|
> |Llama-3.2-3B-Instruct|3.61B|2.11M|0.0586%|
>
> ---
>
> > Q4. **What if the MCTS trace requires more than one loop?**
>
> POLAR by design is not constrained to a single recurrence. Its operation set can be naturally extended to include repeat-2 or repeat-k, which support multiple recurrences. The "at most one recurrence per segment" in our implementation is based on empirical observations of MCTS traces that additional recurrences do not bring noticeable gains (Fig. 7(b) shows effective programs use at most one repetition). Hence, it keeps the program space tractable and avoids unnecessary expansion of the search/prediction space.
>
> This reveals a structural bias of those pretrained LLMs, when viewed as execution-program generators: their training does not further enable a deep, multi-step recurrent usage of their layers. We attribute this behavior to an inductive bias induced by their training objective.
>
>
> ---
>
> We thank the reviewer again for the constructive feedback and supportive assessment, and we will incorporate these clarifications to further improve the presentation and completeness of the paper.

---

> > ### Author Rebuttal · Reviewer_CVa6 · 2026-03-31
> >
> > The rebuttal addressed my questions. Thank you for the helpful explanations and extremely thorough response. I maintain my positive evaluation and hope the paper gets accepted.

---

### Official Review · Reviewer_riB1 · 2026-03-05

**Soundness:** 2
**Presentation:** 2
**Significance:** 2
**Originality:** 2
**Overall Recommendation:** 2
**Confidence:** 4

**Summary:**

The paper proposes viewing inference in LLMs as executing a dynamic program over layers rather than a fixed forward pass. The authors treat each pretrained layer as a reusable function and define an execution program as a sequence of layer calls that can skip or repeat layers. Using MCTS, they show that many inputs admit alternative execution programs, suggesting that fixed-depth inference explores only a limited portion of the model’s latent computation space. To make this practical, they introduce PoLaR, a lightweight predictor that generates input-specific execution programs by segmenting layers and assigning operations (skip, keep, repeat) without modifying the pretrained model weights. Experiments on three reasoning benchmarks show that PoLaR consistently improves accuracy over standard inference and prior dynamic-depth methods.

**Compliance With Llm Reviewing Policy:**

Affirmed.

**Key Questions For Authors:**

- Can you extend the set of evaluation tasks, e.g. GSM8K, MATH, etc?

**Limitations:**

Not provided

**Strengths And Weaknesses:**

Strengths:
- The paper addresses a very important research problem by proposing a unified framework of various dynamic-depth methods such as layer skipping, early exit, recurrence.
- The method does not involve training or fine tuning the model, except training a special router that decides on whether to skip or repeat certain layers.


Weaknesses:
While the paper explores an interesting research direction, the current empirical evidence does not fully convince me of the effectiveness of the proposed approach. I raise several issues and questions below that could help clarify the strengths of the method and address my concerns.

*Analysis on execution programs:*

The paper would benefit from deeper analysis of the MCTS based or learned execution programs. For example, reporting statistics on the average number of skipped layers, which layers are most frequently skipped or repeated, and how often recurrence occurs would help interpret how the method changes the model’s computation and explain the observed gains. It is also not clear whether improvements in Table 3 is coming from early-exit, skipping layers, recurrence or combination of all above.


*On claims:*

In section 2 only subtle variations of the standard model were explored/searched. Based on the Figure 3 it seems like MCTS changes depth between 95% -115% which is only 1-2 layers for LLama-3.2 3B. In the recurrent scenario, they report that MCTS looping over one layer with a single recurrence is effective in most cases. At the same time, paper uses words such as “execution programs become more constrained and increasingly rely on non-trivial execution structures rather than standard inference” or “to be significantly shorter than the standard forward pass”.

*On Evaluation:*

The results in Section 2 in Figure 3 showed very strong improvements. However, they seem to contradict results in Figure 3: Figure 3 shows significant benefits with <95% depth used while in Figure 6 the accuracy improvement on when <100% depth used is marginal.

The OOD evaluation is limited to three benchmarks: ASDiv, MAWPS, and MMLU-Pro. This evaluation is not strong enough to convince claims of improving mathematical reasoning. I strongly recommend increasing the evaluation tasks, at least with GSM8K and MATH.

*Limited upper bound:*

The PoLaR method is based on supervised training from using valid execution programs collected offline via MCTS. Therefore the performance is heavily based on the success of MCTS executing programs and it is the ceiling of the PoLaR approach.


Minor comments on relevant work:

- Authors of [1] have similar motivation on dynamic architecture, but they learn it from scratch.
- In related work on recurrence, authors mention that current approaches demonstrate the value of recurrence, they require architectural redesign and training from scratch. There is the work that addresses this [2] and also discusses the roles of layers.
[1] “Towards Distributed Neural Architectures” (https://arxiv.org/abs/2506.22389)
[2] “Encode, Think, Decode: Scaling test-time reasoning with recursive latent thoughts” (https://arxiv.org/abs/2510.07358)

Minor questions:
- What is the connection to the MoE? If you skip the MoE layer, does it mean skipping every expert?
- What is the empirical evidence of setting K_{max} = 4
- What happens if “repeat” performs >1 additional passes?

---

> ### Author Rebuttal · Authors · 2026-03-31
>
> We thank the reviewer for the detailed and thoughtful feedback, and we address each concern below.
>
> ---
> > Q1. **Analysis on execution programs: The paper would benefit from deeper analysis**
>
> We thank the reviewer for the suggestion. We provide detailed analyses in Fig. R1–R2 and Table R1–R2 ([link](http://anonymous.4open.science/r/review-material-B5CB/Anonymous%20Link%20(ICML%2026).pdf)), and summarize key findings below:
>
> - **MCTS programs.** Skip is roughly uniform across layers, while recurrence is concentrated in middle layers (Fig. R1). Despite an average of ~3-layer reduction in depth, programs exhibit substantial restructuring (avg skip 5.41, avg recur 1.62; Table R1). Without training a decoder per layer, early exit rarely works. Table 1 shows that gains mainly come from the combination of skipping and recurrence rather than early exit alone.
> - **Learned policy.** Skip concentrates on a subset of layers, with recurrence again peaking in middle layers (Fig. R2). Improvements are primarily driven by skipping, with recurrence providing additional gains, while early exit plays a negligible role (Table R2).
>
> Overall, the improvements are not from simple depth reduction, but from structured reorganization of computation.
>
> ---
> > Q2. **On claims: MCTS changes depth between 95%–115% … and recurrence is typically limited … yet the paper claims non-trivial execution structures**
>
> **(1) On “95%–115% depth implies minor changes.”**
>
> Similar *total execution depth* does not imply similar computation.
>
> Total depth counts repeated layer executions and does not reflect computation structure. For example, a program can skip layers and reallocate computation via loops (e.g., skip 4 layers and loop 4 layers), yielding the same depth but a fundamentally different execution path.
>
> Fig. 4 shows that the number of **unique layers can drop to ~80%**, indicating substantial restructuring not captured by total depth.
>
> **(2) On limited recurrence depth.**
>
> While the search space allows deeper recurrence, using more than one additional loop rarely yields valid programs and gains.
>
> This suggests pretrained LLMs favor **shallow, localized reuse** over deep multi-step recurrence, likely due to pretraining inductive bias.
>
> Our claim is not large depth changes, but **input-dependent structural reconfiguration of computation**, which can significantly affect behavior even under similar depth.
>
> ---
> > Q3. **On Evaluation: Figure 3 and Figure 6 seem contradictory …**
>
> Fig. 3 and 6 condition on different quantities and address complementary questions.
> - **Fig. 3 (best-case under a depth budget)**: there often exists a valid execution program with reduced depth that achieves higher accuracy.
> - **Fig. 6 (average over programs at a given depth)**: conditioned on execution depth, average accuracy increases with depth.
>
> Thus, shorter programs suffice for many inputs, while deeper computation improves performance on average.
>
> ---
> > Q4. **On Evaluation: The OOD evaluation is limited …**
>
> We added **GSM8K and MATH500**. The results are consistent with prior OOD findings: **POLAR remains better than the base model and prior dynamic-depth baselines**.
> |Method|GSM8K|MATH500|
> |-|-|-|
> |Base|5.91|6.40|
> |ShortGPT/MindSkip/FlexiDepth|≈ 0|≈ 0|
> |DR.LLM|5.84|5.98|
> |POLAR|**6.12**|**7.80**|
>
> Absolute accuracy is low because base models are **fully frozen, non-math-specialized without CoT**, limiting all methods on these benchmarks.
>
> Importantly, the broader picture is consistent: beyond DART-Math, POLAR already shows **stable OOD gains across ASDiv, MAWPS, and MMLU-Pro**, supporting that the benefit is **not benchmark-specific but reflects generalizable execution control**.
>
> ---
> > Q5. **Limited upper bound: MCTS is the ceiling**
>
> MCTS provides strong training signals rather than a tight ceiling. As shown in Table 1, it yields +30% to +65% gains over standard inference, indicating significant room for improvement in the underlying program space.
>
> Effectively utilizing the discovered program space already yields substantial improvements, and POLAR consistently achieves this without search.
>
> Therefore, the performance is not fundamentally limited by MCTS, but by how well the program space is modeled and exploited.
>
> ---
> > **Minor questions**
> - **MoE:** skipping an MoE layer skips the entire layer (all experts).
> - **$K_{\max}$ = 4:** larger values increase MCTS cost with diminishing returns. Fig. 7(a) shows a strong bias toward short segments (54.5% length 1), so small $K_{\max}$ captures dominant patterns.
> - **Repeat >1:** We investigated this via MCTS and consistently observed that valid programs use at most a single repetition (Fig. 7(b)), motivating our design; see response to Q4 (Reviewer CVa6) for further discussion.
> - **Related work:** We thank the reviewer and will include these works and clarify their relation.
>
> ---
> We hope these clarifications address the reviewer’s concerns and better demonstrate the effectiveness and interpretability of POLAR.

---

> > ### Author Rebuttal · Reviewer_riB1 · 2026-04-02
> >
> > Thanks for your work and providing additional information.
> >
> > > Re: On claims and feedback to statistical analysis of PoLaR and MCTS
> > Thanks for the statistics of skipping/looping for MCTS and PoLaR.
> > - Interestingly, they have very different distributions. Although MCTS was guiding PoLaR during training. It raises how the observations based on MCTS transfer to PoLaR method.
> > - The statistics of PoLaR for skipping/looping supports previous research on layer-to-layer functionality.
> >
> > I think Figure R2 and Table R2 support my previous comments that in the end the proposed “input-dependent structural reconfiguration of computation” leads to *minor changes to the general architecture with minor gains*.
> >
> > Figure 4 reports execution depth of the MCTS approach and based on the Figure R1 vs R2 MCTS and PoLaR have different behaviour.
> >
> > I am curious how *random limited skipping or looping over one layer* will impact performance. In this experiment, one should control that random skipping or looping cannot be significant, i.e. skipping on average 2-3 layers and looping 2-3 times.
> >
> > > Re: evaluation
> >
> > Thanks for clarifying that  Fig. 3 presents best-case and Fig. 6 average-case results. Although it is not clear from just reviewing the figures standalone, I acknowledge it was provided in the main text.
> >
> > The results on new benchmarks (GSM8K, MATH) are marginal. Authors acknowledge that absolute performance is low due to the base capabilities of the models. Then it makes it difficult to evaluate the method based on new results, which does not make evaluations stronger than it was before.
> >
> >
> > In conclusion, I think research problem is very interesting and important. However, based on all evidence including additional one, I think that the PoLaR method, i.e. method for input-dependent structural reconfiguration of computation, leads to small reconfiguration of computation and small performance improvements.
> >
> > I will maintain my score.

---

> > > ### Author Response · Authors · 2026-04-03
> > >
> > > We thank the reviewer for the constructive follow-up. The following additional clarification and results are to address the remaining concerns.
> > >
> > > ---
> > >
> > > ### 1. On the gap between MCTS and PoLaR behaviors
> > >
> > > The difference in output distribution between MCTS (Fig. R1) and PoLaR (Fig. R2) is expected but it does not affect the effectiveness of PoLaR.
> > >
> > > MCTS discovers multiple valid programs per sample in a larger program space, whereas the training of PoLaR captures the most reliable patterns reusable across different samples. This simplifies the predicted programs and leads to better generalization. PoLaR does not aim to match the full distribution of MCTS. Instead, it aims to maintain the performance gains of MCTS programs.
> > >
> > > ---
> > >
> > > ### 2. On whether simple/random modifications can explain gains
> > >
> > > We tested this using simple random baselines (4 models × 5 difficulty levels × 3 strategies, 60 settings in total): randomly skipping 2–3 layers, looping 2–3 layers, or both.
> > >
> > > All random variants achieve **~0% accuracy (≤0.2%)**.
> > >
> > > This implies that effective programs lie in an **extremely sparse region** of the combinatorial space of programs. The observed gains on our programs are **not due to small or random perturbations**, but require **structured, input-dependent organization of layer-skip and recurrence**, which is achieved by PoLaR.
> > >
> > > ---
> > >
> > >
> > > ### 3. On GSM8K / MATH results being marginal
> > >
> > > The poor performance on the requested benchmarks is expected, as neither the base model nor PoLaR is trained for these reasoning tasks, which usually require long reasoning traces. Any post-training is bottlenecked by the base capability.
> > >
> > > Despite this, PoLaR still **consistently improves over the frozen base model**. In more feasible settings, PoLaR achieves **substantial in-distribution gains (up to +20.8%)** and **consistent OOD improvements over multiple benchmarks**.
> > >
> > > ---
> > >
> > > We respectfully clarify that we do not claim **large architectural change** but advocate
> > >
> > > > **structured, input-adaptive programs that lead to consistent, non-trivial gains**
> > >
> > > which random or trivial modifications cannot achieve.

---

### Official Review · Reviewer_kZ2o · 2026-03-09

**Soundness:** 3
**Presentation:** 3
**Significance:** 2
**Originality:** 2
**Overall Recommendation:** 4
**Confidence:** 2

**Summary:**

This paper treats each layer of pretrained LLMs as an execution unit, and studies how to program the layers via skip, keep, repeat conditioned on input to improve the performance. The authors first use Monte-Carlo Tree search to explore the possibility of shorter/better program for each input, and have several empirical observations: combining layer skipping and recurrence yields better performance; many valid programs are shorter than the standard forward pass; increasing execution complexity improves performance on harder inputs; Valid programs typically operate on short contiguous layer segments and involve at most one recurrence. Based on these observations, they further propose POLAR that trains a separate network on the MCTS data to predict the program of layers given input. Experiments show that POLAR has better performance on reasoning benchmarks than standard and other adaptive layer method, and POLAR can generalize to OOD benchmarks.

**Compliance With Llm Reviewing Policy:**

Affirmed.

**Final Justification:**

The rebuttal adequately addressed my concern. I maintain my evaluation for accepting.

**Key Questions For Authors:**

1. How computationally expensive POLAR is, including using MCTS to generate the supervision data and training the prediction network?
2. How would the authors predict the performance of POLAR compare to looped architecture which is converted from pretrained model e.g. McLeish et al. , 2025, Teaching Pretrained Language Models to Think Deeper with Retrofitted Recurrence?

**Limitations:**

yes

**Strengths And Weaknesses:**

Strengths:
1. Extensive experiments are done to confirm the benefit of layer skipping and layer recurrence and study what kind of program of layers can improve the performance.
2. The design choices of POLAR are well-justified by the empirical observations e.g. using short contiguous layer segments and recurrence at most once to reduce the search space.
3. Superior pass@k performance than other related methods and good OOD performance e.g. history, health, etc.

Weaknesses:
1. Related work mentions Li et al. (2025) and (Heakl et al., 2025), but does not mention Li et al. (2025) already uses MCTS to find the program of layers and (Heakl et al., 2025) already uses the data generated by MCTS to train a prediction network. In other words, from only  reading the paper, I think this paper first uses the MCTS to find program of layers.
2. Some of the main observations (combining layer skipping and recurrence yields better performance; many valid programs are shorter than the standard forward pass) are already mentioned by Li et al. (2025).

Since I'm not an expert in this field, I'm not sure how to evaluate the significance of the results. Apologies in advance.

---

> ### Author Rebuttal · Authors · 2026-03-31
>
> We thank the reviewer for the thoughtful and constructive feedback. We are encouraged that the reviewer recognizes the strength of our empirical findings and the consistent OOD generalization, which we believe are key indicators of the effectiveness and practical value of our approach.
>
> We address the questions below.
>
> ---
> > Q1. **… not the first uses the MCTS to find program of layers …**
>
> We thank the reviewer for the comment and would like to clarify a potential misunderstanding.
>
> **MCTS is not a technical contribution claimed in our work.** As a widely-used search strategy, we use it for **diagnosis and analysis (which are novel)** of program-of-layers:
> > “We use MCTS strictly as a diagnostic tool…”
>
> Our phrasing “first empirical study of its kind” may have been ambiguous. Here, “first” refers to our **systematic and empirical characterization of the program-of-layers space**, rather than the use of MCTS itself.
>
> Our key contribution is to **translate the insights revealed by MCTS into design principles**, and further into a **practical method (POLAR)** that replaces MCTS with end-to-end program prediction.
>
> ---
> > Q2. **Some observations are already mentioned by Li et al. (2025)**
>
> We agree that Li et al. (2025) shares some high-level observations (e.g., the benefit of skip and recurrence). However, they primarily focus on **using MCTS to discover better execution paths**, and report these observations as empirical findings.
>
> In contrast, we present a **systematic characterization of the program-of-layers space**, which reveals several novel structural properties, e.g., *segment-level structure*, *prevalence of shorter programs*, and *scaling behavior*. These properties are crucial to **motivate the design of our learning-based method (POLAR)**.
>
> Therefore, compared to Li et al. (2025), **our analysis establishes a structured understanding of the space and translates it into a practical inference framework**.
>
> ---
> > Q3. **How computationally expensive POLAR is, including using MCTS to generate the supervision data and training the prediction network?**
>
> The computation in POLAR is composed of three parts: **offline data generation, lightweight training, and efficient inference**.
>
> - **Offline (MCTS data generation).**
>   MCTS is used *only once* to construct supervision data and is **not part of inference-time computation**. While it introduces additional offline cost, this is for one-time training that saves the search cost at deployment time.
> - **Training (predictor).**
>   The learned predictor is **extremely lightweight (~2.1M parameters)**, corresponding to only **0.01%–0.06% of the base LLM size** (e.g., 2.11M vs 14.32B). This small footprint is consistent across model scales (3B–14B), making training **both simple and inexpensive**, negligible compared to standard LLM pretraining or fine-tuning (see CVa6 Q3).
> - **Inference (POLAR predictor).**
>   At inference, POLAR replaces MCTS with the lightweight predictor and a small beam search (no per-input search). The total overhead is **~0.8% of a forward pass (<1 LLM layer)** (see CVa6 Q2), making it negligible in practice. Importantly, by reducing execution depth, POLAR often **reduces end-to-end latency** (0.83× on easy & 0.95× on hard inputs) while improving accuracy.
>
> **In summary:** POLAR reduces costly inference-time search to one-time offline training, requires **minimal training cost**, adds **negligible inference overhead, and reduces inference depth, making it faster than the base model while improving performance**.
>
> ---
>
> > **Q4. Comparison to looped architectures (e.g., McLeish et al., 2025)**
>
> Looped architectures require **architectural modification and retraining** to introduce recurrence. In contrast, **POLAR operates on frozen pretrained LLMs and performs inference-time program selection** (skip/repeat) without retraining.
>
> This reflects a fundamental difference: looped methods **learn new computation mechanisms**, while POLAR **exploits the latent computation already present in pretrained models**. Our empirical analysis shows that pretrained LLMs admit many alternative valid execution programs (often shorter or more accurate than the standard forward pass), indicating substantial unexplored computational potential that POLAR can activate at inference time.
>
> ---
>
> We hope these clarifications address the reviewer’s concerns and better highlight the novelty and significance of our work.

---

> > ### Author Rebuttal · Reviewer_kZ2o · 2026-04-01
> >
> > Can you elaborate more about the methodology difference between this paper and Heakl et al., 2025? I'm leaning toward accepting this paper if the author can be clearer about the contribution of prior work (e.g. Heakl et al., 2025, Li et al., 2025 etc) and how this work differ from them in the camera-ready version.

---

> > > ### Author Response · Authors · 2026-04-02
> > >
> > > We thank the reviewer for the suggestion. We will revise the camera-ready to present these distinctions more clearly and concisely.
> > >
> > > ---
> > >
> > > ### **Difference from Heakl et al. (2025 / Dr.LLM)**
> > >
> > > We emphasize two key differences:
> > >
> > > **(1) Model design (efficiency of decision making).**
> > > Dr.LLM performs *sequential, layer-wise routing*, where each decision depends on previous layer's hidden states during the forward pass. In contrast, our method predicts the **entire execution program upfront, before any forward computation**, eliminating interleaved routing and execution overhead. This leads to a fundamentally more efficient inference paradigm.
> > >
> > > **(2) Program space (strictly more expressive).**
> > > Our method operates over a **strictly larger program space** than Dr.LLM. Specifically, Dr.LLM is limited to **single-layer recurrence** (e.g., 4→4), while our method can **encapsulate multiple layers into a recurrent unit** (e.g., 4→5→4→5). Therefore, Dr.LLM’s program space is a strict subset of ours, and our method can generate various programs that Dr.LLM cannot. This additional expressiveness translates to substantial advantage: **45.5% of learned programs involve multi-layer recurrent structures (Fig. 7(a))**, which are **inexpressible in Dr.LLM’s design**. This gap in representational capacity is reflected in the final performance improvements (Table 2 and 3).
> > >
> > > ---
> > >
> > > ### **Difference from Li et al. (2025)**
> > >
> > > **(1) From analysis to a practical method.**
> > > Li et al. (2025) stop at MCTS-based analysis, which is not practical, as it requires access to ground-truth outputs and incurs approximately 100× additional forward passes. In contrast, we use MCTS as a diagnostic tool, and the resulting insights motivate us to develop a practical method that reduces overall inference cost on average.
> > >
> > > **(2) From program discovery to structural understanding.**
> > > Li et al. focus on discovering better execution programs via search, whereas our work aims to understand the **structure of the program space** and leverage it for model design (response to Q2 in the initial rebuttal).

---

### Official Review · Reviewer_HeVk · 2026-03-10

**Soundness:** 4
**Presentation:** 4
**Significance:** 4
**Originality:** 3
**Overall Recommendation:** 5
**Confidence:** 3

**Summary:**

This paper introduces POLAR (Program-of-Layers), a framework that interprets the layers of a large language model (LLM) as a library of atomic functions. Instead of the standard fixed-depth forward pass, POLAR dynamically predicts an input-specific execution path (of skipping or repeating layers) to optimize accuracy and efficiency without modifying the pretrained model's parameters. Unlike previous dynamic inference methods that typically restrict execution to a single operation type, such as only skipping layers for speed or only looping for reasoning, POLAR optimizes in a joint space of both skipping and recurrence.

**Compliance With Llm Reviewing Policy:**

Affirmed.

**Final Justification:**

The paper offers a neat and economical method to boost transformer-based LLMs in terms of both accuracy and runtime. I was mostly concerned with the potential overhead of the POLAR prediction network and the MCTS costs for data generation, but both were addressed in the authors' rebuttal. Therefore, I maintain my recommendation for acceptance.

**Key Questions For Authors:**

1. Does the computational cost of running the POLAR prediction network (which uses a cross-layer encoder and beam search) negate the latency benefits gained by skipping layers on easier tasks?
2. The segment length was capped at **K_max = 4** based on empirical evidence. Was a dynamic segmentation approach (where the network predicts segment length freely) tested, and did it fail due to training instability or search space explosion?
3. Based on the current scheme, the framework can loop a segment for at most two iterations via the *repeat* operation. Have you considered extending this by supporting a variable number of loops?
4. Do you have a deeper explanation for the out-of-domain generalization of the method? It suggests that programs-of-layers operate in a space that is largely agnostic to task specifics; what is the underlying hypothesis for this structural inductive bias?

**Limitations:**

yes

**Strengths And Weaknesses:**

# Strengths
1. **Unified "Skip and Loop" Framework:** Unlike prior works that focus exclusively on either layer skipping or layer looping, POLAR successfully unifies both into a single framework. The authors empirically demonstrate that combining "skip" and "loop" operations yields higher accuracy than either method in isolation.
2. **Frozen Pretrained Models:** A significant practical advantage is that POLAR operates on fully frozen pretrained models. It optimizes inference by learning a lightweight prediction network rather than requiring expensive fine-tuning or architectural modifications to the LLM itself.
3. **Strong Out-of-Distribution Generalization:** The model demonstrates impressive out-of-distribution capabilities. Despite being trained on mathematical reasoning data (DART-Math), the learned execution programs improved performance on diverse MMLU-Pro subjects like Law, History, and Biology. This supports the claim that the model learns transferable computation control strategies rather than dataset-specific heuristics.
4. **Rigorous Validation via MCTS:** The authors utilize Monte Carlo Tree Search (MCTS) not just as a distinct method, but as a diagnostic tool to validate the existence of better latent execution paths before training their predictor. This confirms that standard forward passes often over-compute on easy tasks and under-compute on hard ones.
5. **Effective Test-Time Scaling:** The paper provides evidence of effective test-time scaling. Increasing the number of candidate execution programs (**k**) monotonically improves accuracy, allowing the system to trade compute for accuracy dynamically.
# Weaknesses

1. **Training Scalability Bottleneck:** While the inference phase uses a lightweight predictor, the training of that predictor relies on supervision from valid execution programs collected offline via MCTS. Generating this ground truth data is computationally expensive and explores an exponentially large, non-convex program space, which may hinder scalability when preparing training data for very large datasets.
2. **Missing Wall-Clock Latency Evaluation:** The evaluation focuses heavily on **Pass@k** accuracy and the number of unique layers executed. While the paper mentions that shorter programs are preferred, there is no explicit analysis of wall-clock latency that accounts for the computational overhead of the POLAR prediction network and the beam search used during decoding.

---

> ### Author Rebuttal · Authors · 2026-03-31
>
> We appreciate the reviewer’s highly positive and insightful feedback. We are especially encouraged that the reviewer recognizes POLAR as a principled step toward *adaptive computation in pretrained LLMs*, enabling input-dependent execution through a unified and flexible program over layers without modifying model weights.
>
> We address the questions below.
>
> ---
> > Q1. **Training scalability bottleneck due to MCTS supervision**
>
> Our method does **not require extensive search** but only *sufficiently good valid programs* for training.
>
> Although the program space is large, there exist many programs that lead to correct answers. Empirically, **useful programs emerge earlier in exploration**: with only **100 MCTS simulations**, we can collect sufficient **diverse valid programs** (up to **39.4 per sample on average**).
>
> This small budget already suffices to train a strong programmer: **Table 1 shows MCTS improves over standard inference by +30%–65%**, indicating effective exploration of a rich program space. Increasing the budget to 200 yields **no significant gains**, further confirming diminishing returns.
>
> ---
> > Q2. **Does the computational cost of running the POLAR prediction network … negate the latency benefits gained by skipping layers?**
>
> POLAR’s inference cost includes (i) a lightweight predictor and (ii) a small beam search over a highly constrained space.
>
> Latency breakdown (Qwen1.5-MoE-A2.7B-Chat, 24 layers):
>
> **Table X. Component-wise inference overhead in POLAR.**
> |Component|Latency (ms)|Equivalent LLM layers|% of full forward|
> |-|-:|-:|-:|
> |1 LLM layer|13.23|1.0|3.5%|
> |Predictor head| 0.99 |0.1|0.3%|
> |Beam search|0.11|0.01|0.03%|
> |**Predictor + beam**|**1.10**|**0.1**|**0.3%**|
> |Encoder (amortized)|1.95|0.1|0.5%|
> |**Total overhead**|**3.05**|**0.2**|**0.8%**|
>
> Thus, POLAR's inference overhead (even including the encoder) is **< 1 LLM layer**.
>
> **Table Y. End-to-end latency, execution depth, and accuracy.**
> |Method|Avg. layers|Latency (ms)|Rel. to Base|Accuracy (Δ vs Base)|
> |-|-:|-:|-:|-:|
> |Base (24 layers)|24.00|373.45|1.00x|—|
> |POLAR (DM-1, easier inputs)|23.30|311.41|0.83x|**+5.8**|
> |POLAR (DM-5, harder inputs)|23.76|353.31|0.95x|**+1.2**|
>
> These results show POLAR achieves higher accuracy with lower latency (0.83× on easy & 0.95× on hard inputs).
>
> ---
> > Q3. **K_max = 4 and dynamic segmentation**
>
> Empirically, the effective program space is highly concentrated on short, local segments. As shown in Fig. 7(a), 54.5% of recurrences have length 1, while longer segments occur much less frequently.
>
> This structural bias suggests that restricting the segment length ($K_{\max}=4$) suffices to cover most valid programs in practice. Therefore, our constraint does not reduce the effective program space but instead improves efficiency significantly by constraining an otherwise combinatorial search space.
>
> Allowing fully unconstrained segmentation would substantially increase the complexity of both search and learning, with marginal empirical benefits.
>
> ---
> > Q4. **Extending to variable number of loops**
>
> POLAR by design is not constrained to a single recurrence. Its operation set can be naturally extended to include repeat-2 or repeat-k, which support multiple recurrences. The "at most one recurrence per segment" in our implementation is based on empirical observations of MCTS traces that additional recurrences do not bring noticeable gains (Fig. 7(b) shows effective programs use at most one repetition). Hence, it keeps the program space tractable and avoids unnecessary expansion of the search/prediction space.
>
> This reveals a structural bias of those pretrained LLMs, when viewed as execution-program generators: their training does not further enable a deep, multi-step recurrent usage of their layers. We attribute this behavior to an inductive bias induced by their training objective.
>
> ---
> > Q5. **Do you have a deeper explanation for the out-of-domain generalization?**
>
> While the OOD generalization indicates that the POLAR prediction head itself is domain-agnostic (trained on math but generalizing to others), the whole process of program prediction is not, because the external embedding model was trained across diverse domains. It maps inputs to a domain-agnostic representation space where the prediction head trained on single domain can generalize. Note the prediction head is much smaller than the embedding model. Hence, the program prediction still leverages the domain information.
>
> ---
> We thank the reviewer again for the constructive feedback and supportive assessment, and we will incorporate these clarifications to further strengthen the paper.

---

> > ### Author Rebuttal · Reviewer_HeVk · 2026-04-03
> >
> > Thank you for your thorough rebuttal. I find my main concerns adequately addressed.
> >
> > As such, I maintain my positive score, and recommend that the authors add the details provided in this rebuttal to the final manuscript.
> >
> > Great work!

---

### Decision · Program_Chairs · 2026-04-30

**Decision:**

Accept (spotlight)

**Comment:**

This paper introduces POLAR (Program-of-Layers), a highly original framework that treats pretrained LLM layers as a library of atomic functions. By utilizing a lightweight predictor to dynamically determine an execution path (skipping or repeating layers) for each input, it achieves adaptive computation without requiring expensive fine-tuning or architectural modifications to the base model.

The reviewers broadly praised the paper for its unified "skip and loop" approach and its insightful use of Monte Carlo Tree Search (MCTS) as a diagnostic tool to map the latent computation space. The empirical evaluation is robust, demonstrating improved accuracy, reduced latency, and impressive out-of-distribution generalization.

While Reviewer riB1 maintained a negative score—arguing that the method results in only minor architectural changes and marginal gains—the consensus among the other reviewers and the Area Chair is that this assessment overlooks the core contribution. The authors successfully demonstrated in their rebuttal that POLAR's gains stem from a structured, input-dependent reorganization of computation, which cannot be replicated by random or trivial skip/loop modifications. The paper's approach to activating latent computational pathways in frozen models is a significant step forward for the field.

This is a technically solid, innovative paper that makes a meaningful contribution to adaptive computation in LLMs.